# IGU-LoRA: Adaptive Rank Allocation via Integrated Gradients and Uncertainty-Aware Scoring

**Xuan Cui**[1], **Huiyue Li**[1], **Run Zeng**[1], **Yunfei Zhao**[1], **Jinrui Qian**[1], **Wei Duan**[1,*], **Bo Liu**[1,*], **Zhanpeng Zhou**[2,*]
[1]Chongqing Technology and Business University    [2]Shanghai Jiao Tong University
* Corresponding Author

## Abstract

As large language models (LLMs) scale to billions of parameters, full-parameter fine-tuning becomes compute- and memory-prohibitive. Parameter-efficient fine-tuning (PEFT) mitigates this issue by updating only a small set of task-specific parameters while keeping the base model frozen. Among PEFT approaches, low-rank adaptation (LoRA) is widely adopted; however, it enforces a uniform rank across layers despite substantial variation in layer importance, motivating layerwise rank allocation. Recent adaptive-rank variants (e.g., AdaLoRA) allocate ranks based on importance scores, yet typically rely on instantaneous gradients that capture only local sensitivity, overlooking non-local, pathwise effects within the same layer, which yields unstable and biased scores. To address this limitation, we introduce **IGU-LoRA**, an adaptive-rank LoRA that (i) computes within-layer Integrated Gradients (IG) sensitivities and aggregates them into a layer-level score for rank allocation, and (ii) applies an uncertainty-aware scheme using exponential moving averages with deviation tracking to suppress noisy updates and calibrate rank selection. Theoretically, we prove an upper bound on the composite trapezoidal rule approximation error for parameter-space IG under a pathwise Hessian-Lipschitz condition, which informs the quadrature budget. Across diverse tasks and architectures, IGU-LoRA consistently outperforms strong PEFT baselines at matched parameter budgets, improving downstream accuracy and robustness. Ablations confirm the contributions of pathwise within-layer sensitivity estimates and uncertainty-aware selection to effective rank allocation. Our code is publicly available at https://github.com/withyou12/igulora.git

## 1 Introduction

Large language models (LLMs) have achieved remarkable success across a wide range of NLP tasks (Devlin et al., 2019; Brown et al., 2020a; Han et al., 2025). However, specialising these models for new downstream tasks remains challenging due to their large parameter counts and substantial computational and memory costs. Consequently, fine-tuning has emerged as the standard way to adapt pre-trained LLMs to particular downstream tasks.

Early efforts in fine-tuning primarily relied on full-parameter fine-tuning (FPFT) (Lv et al., 2024; Qiu et al., 2020; Raffel et al., 2020), where all model parameters are updated during training. While effective for small to medium-scale models, such as BERT (Devlin et al., 2019) and RoBERTa-large (Liu et al., 2019), FPFT becomes increasingly impractical as model size scales exponentially. For example, GPT-3 (Brown et al., 2020b) contains 175 billion parameters, making full fine-tuning prohibitively expensive in terms of computation and memory.

To alleviate these challenges, parameter-efficient fine-tuning (PEFT) methods have been proposed, which adapt pre-trained models by updating only a small subset of parameters while keeping most of the model frozen. Notable PEFT methods include Adapter Tuning (Houlsby et al., 2019; Rücklé et al., 2021; Pfeiffer et al., 2021; He et al., 2022; Wang et al., 2022), Prefix Tuning (Li & Liang, 2021; Wu et al., 2024), and Prompt Tuning (Liu et al., 2022b; Zhang et al., 2024; Yu et al., 2023; Cui et al., 2025). These methods significantly reduce the number of trainable parameters. However, they primarily affect shallow or intermediate layers, limiting their ability to capture deeper semantic representations.

Complementary to the above, weight-delta methods (e.g., Diff Pruning (Guo et al., 2020; Fang et al., 2023)) selectively update a sparse subset of important weights. While effective in reducing the scale of trainable parameters, these methods often rely on unstructured sparsity, which poses challenges for optimisation and is less compatible with modern hardware acceleration. A more structured alternative is Low-Rank Adaptation (LoRA) (Hu et al., 2022a), which models the weight update $\mathbf{\Delta W}$ as the product of two low-rank matrices. By preserving the pretrained model architecture and introducing only a small number of trainable parameters, LoRA achieves high efficiency without sacrificing model capacity. However, LoRA typically uses a fixed rank across all layers, ignoring the heterogeneous contribution of different weight matrices. This static configuration may limit the adaptability and expressiveness of the model.

Building on this observation, several adaptive-rank PEFT methods have been proposed (Zhang et al., 2023; Xu et al., 2023; Ding et al., 2023; Valipour et al., 2023). For example, AdaLoRA (Zhang et al., 2023) applies singular value decomposition (SVD) to the low-rank update matrices and dynamically adjusts rank sizes based on layer-wise importance scores. However, the scoring mechanism in AdaLoRA is primarily based on instantaneous gradient signals, which fail to capture long-term parameter contributions and inter-layer interactions. As a result, the rank allocation may be suboptimal in complex optimisation scenarios.

To overcome these limitations, we propose IGU-LoRA(Fig. 1(c)), an IG-driven PEFT framework that extends Integrated Gradients to the parameter space for scoring parameter importance. The IG path integral is efficiently approximated via a mini-batch stochastic quadrature that uniformly samples one node $\alpha \in [0, 1]$ per mini-batch, thereby avoiding the $O(N)$ forward-backward passes of trapezoidal integration—where $N$ denotes the number of discretization steps along the IG path—and adding only batch-linear overhead. Compared with instantaneous-gradient heuristics, this yields stable and globally informed importance estimates. Robustness is further enhanced by modeling the predictive effect of parameter perturbations and by an uncertainty-aware score that couples an EMA mean with a dispersion term. On the theory side, we establish (i) a discretization-sampling error bound for the IG estimator of order $O(N^{-2}) + O(M^{-1/2})$, where $M$ is the number of sampled mini-batches, and (ii) a high-probability stability guarantee for the EMA ratio score $\mathsf{SNR}t$, the signal-to-noise ratio at iteration $t$. Empirically, across datasets (BoolQ, GSM8K, GLUE, ...) and backbones (RoBERTa-large, Qwen-2.5-0.5B, Llama-2-7B, Llama-3-8B, DeepSeek-R1-Distill-Qwen-2.5-7B), IGU-LoRA consistently improves accuracy over strong PEFT baselines (LoRA, AdaLoRA, DoRA) while matching their memory footprint and decoding latency.

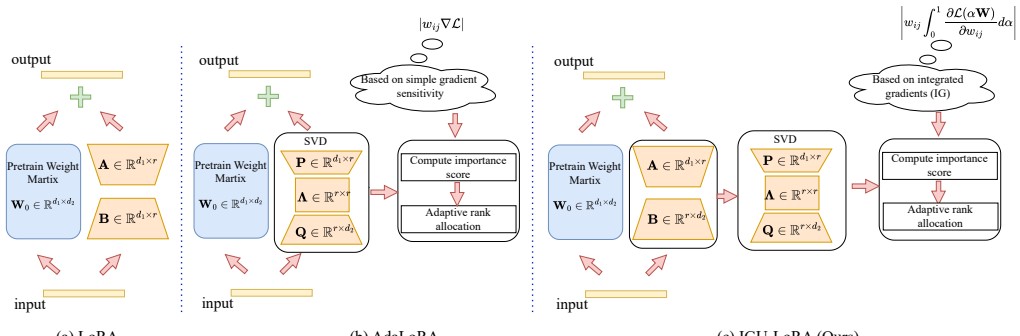

**Figure 1:** Comparison of frameworks: left to right—(a) LoRA, (b) AdaLoRA, (c) the proposed IGU-LoRA. IGU-LoRA builds on LoRA and AdaLoRA, introducing integrated gradients (IG) to compute parameter importance scores. Please zoom in 300% for better clarity.

## 2 RELATED WORKS

### 2.1 PARAMETER EFFICIENT FINE-TUNING

Parameter-Efficient Fine-Tuning (PEFT) received widespread attention for its effectiveness in efficiently adapting LLMs. Representative approaches included Adapter Tuning (Houlsby et al., 2019; Rücklé et al., 2021; Pfeiffer et al., 2021; He et al., 2022; Wang et al., 2022), Prefix Tuning (Li & Liang, 2021; Wu et al., 2024), Prompt Tuning (Liu et al., 2022b; Zhang et al., 2024; Yu et al., 2023; Cui et al., 2025), and P-Tuning v2 (Liu et al., 2021), which inserted lightweight trainable modules into different layers of the model to enable efficient task adaptation. In parallel, reparameterization-based PEFT approaches (Li et al., 2018; Aghajanyan et al., 2021; Liu et al., 2024a; Hu et al., 2022a; Zhang

et al., 2023) received increasing attention. Without modifying the model architecture, these methods modeled and optimized parameter updates in a low-dimensional and efficient manner. Among them, Low-Rank Adaptation (LoRA) (Hu et al., 2022a) has become a prominent method by expressing weight updates as the product of two low-rank matrices, which allows for tight control over the trainable parameter count while maintaining model performance. With the rapid release of open-source LLMs (Shao et al., 2024; Liu et al., 2019; Dubey et al., 2024) and their increasing use in instruction tuning and other real-world applications, PEFT has emerged as the mainstream paradigm for scalable fine-tuning and has been widely adopted in practical systems.

## 2.2 LOW-RANK ADAPTATION FINE-TUNING

LoRA (Hu et al., 2022a) is a representative PEFT method that freezes pretrained weights and injects low-rank matrices, reducing parameter overhead with minimal performance loss. Several LoRA-based methods have been proposed to enhance efficiency and scalability. For example, Delta-LoRA (Zi et al., 2023) improves LoRA's expressiveness by updating weights with the temporal difference of $\mathbf{AB}$, addressing the limitations of small low-rank matrices. DoRA (Liu et al., 2024b) decouples optimization by factorizing $\mathbf{W}$ into a magnitude vector $m$ and a direction matrix $\mathbf{V}$. MeLoRA (Ren et al., 2024) aggregates outputs from parallel low-rank adapters in a block-diagonal structure to improve model capacity. AutoLoRA (Xu et al., 2023) uses meta-learning to automatically assign optimal per-layer ranks, while AdaLoRA (Zhang et al., 2023) dynamically adjusts ranks during training using SVD and parameter importance scores. SalientLoRA (Ke et al., 2024) allocates ranks based on parameter saliency, optimizing the low-rank layers for improved performance. GoRA (He et al., 2025) adapts low-rank adjustments dynamically using gradient-driven methods to meet task requirements while maintaining efficiency. These techniques enable efficient fine-tuning with fewer trainable parameters and strong performance.

## 2.3 INTEGRATED GRADIENTS

In interpretability research for deep learning, Integrated Gradients (IG (Sundararajan et al., 2017)) is a widely adopted attribution method that mitigates gradient saturation by computing the integral of gradients along the path from a baseline input to the actual input. IG satisfies two key axioms, completeness and sensitivity, which ensure that it quantitatively reflects the contribution of each input feature to the model's output. Subsequent studies extend IG in various directions. Theoretically, Lundberg & Lee (2017) show that IG is equivalent to Shapley values under certain conditions. From a computational perspective, Kapishnikov et al. (2021) propose an adaptive sampling strategy that improves runtime efficiency by $3\times$. IG also demonstrates practical utility in high-stakes domains such as medical imaging (Sayres et al., 2019), where it improves the localization of diabetic retinopathy markers. In this work, we extend IG to parameter importance estimation in large model fine-tuning. Our method addresses the limitations of instantaneous gradient signals, which are prone to vanishing in deep networks. It introduces a redefined sensitivity scoring mechanism that more accurately captures long-term parameter contributions during optimization.

## 3 METHOD

### 3.1 PRELIMINARIES

**Low-Rank Adaptation.** Low-Rank Adaptation (LoRA (Hu et al., 2022a)) injected trainable low-rank matrices into frozen pre-trained weights, substantially reducing the number of trainable parameters while preserving downstream task performance. Given a pre-trained parameter matrix $\mathbf{W}_0 \in \mathbb{R}^{d_1 \times d_2}$ for a specific layer of an LLM, LoRA updated the parameter matrix as:

$$\mathbf{W} = \mathbf{W}_0 + \mathbf{AB}, \tag{1}$$

where $\mathbf{A} \in \mathbb{R}^{d_1 \times r}$ and $\mathbf{B} \in \mathbb{R}^{r \times d_2}$ were low-rank trainable matrices with $r \ll \min\{d_1, d_2\}$.

**Adaptive LoRA.** A key limitation of LoRA is that it requires manually selecting the rank $r$, which is challenging due to the heterogeneity of intrinsic dimensionalities across layers and the lack of principled guidance for determining appropriate values. To enable adaptive rank selection, singular value decomposition (SVD) is typically applied to the trainable low-rank product $\mathbf{AB}$ in Eq. (1) (Zhang et al., 2023):

$$\mathbf{W} = \mathbf{W}_0 + \text{SVD}(\mathbf{AB}) = \mathbf{W}_0 + \mathbf{P}\Lambda\mathbf{Q}, \tag{2}$$

where $\mathbf{P} \in \mathbb{R}^{d_1 \times r}$, $\mathbf{Q} \in \mathbb{R}^{r \times d_2}$ are two orthogonal matrices, and the diagonal matrix $\Lambda = diag\{\lambda_1, \lambda_2, \ldots, \lambda_r\} \in \mathbb{R}^{r \times r}$ containing the singular values. We initialize $r$ as an overparameterized upper bound $r \ll \min\{d_1, d_2\}$, then prune redundant dimensions via spectral analysis.

To determine the final rank, we define an importance score $S_i$ for each singular value $\lambda_i$, which guides the pruning process. Unlike conventional methods that rely solely on magnitude, our proposed scoring method incorporates both the singular value and the sensitivity of its associated parameters, namely the elements in the $i$-th column of $\mathbf{P}$ and the $i$-th row of $\mathbf{Q}$. Specifically, for each $i \in \{1, \ldots, r\}$, we estimate $S_i$ by aggregating two components. First, $s_\lambda(\cdot)$ measures the intrinsic strength of the singular value; Second, $s_e(\cdot)$ quantifies the importance of the parameters with the $i$-th column of matrix $\mathbf{P}$ and the $i$-th row of matrix $\mathbf{Q}$. The final score $S_i$ is computed as Zhang et al. (2023):

$$S_i = s_\lambda(\lambda_i) + \frac{1}{d_1}\sum_{k=1}^{d_1} s_{snr}(P_{ki}) + \frac{1}{d_2}\sum_{k=1}^{d_2} s_{snr}(Q_{ik}), \qquad (3)$$

where $s_\lambda(\lambda_i) = |\lambda_i|$ denotes the magnitude of the singular value, and $s_{snr}(\cdot)$ is a specific importance score function that measures the importance of individual weight on the training loss function. Existing methods (Zhang et al., 2023) for measuring parameter importance are primarily based on simple gradient sensitivity $|w_{ij}\nabla_{w_{ij}}\mathcal{L}|$, where $w_{ij}$ is a single parameter in model. However, this simple gradient sensitivity-based method suffers from the following limitations:

• **Lack of Structural Interpretability:** Simple gradient sensitivity-based method evaluate weights independently, ignoring the structured interactions among parameter groups. In settings like LoRA, where parameters operate collectively within subspaces, such element-wise assessments fail to capture their joint contribution, thereby limiting interpretability at the structural level.

• **Instantaneous Parameter Sensitivity:** Simple gradient sensitivity-based method capture only the instantaneous impact of a parameter on the loss function, overlooking its accumulated or long-term contribution throughout training. This limitation can result in unstable or misleading estimates.

• **Gradient Saturation:** In transformer-based LLMs, activation functions such as ReLU may lead to gradient saturation in inactive regions, where the gradient signal vanishes entirely. As a result, the estimated importance of the affected parameters becomes unreliable.

Figure 2 illustrates why (a) the simple gradient method fails in gradient-saturated regions, while (b) the integrated gradient method provides more reliable parameter importance estimation through a comparative demonstration. To address these limitations, we estimate parameter importance using Integrated Gradients (IG) in the parameter space. IG integrates the gradients along the path from 0 to 1, thereby capturing the non-local sensitivity and overall impact of the gradients. This method not only accounts for the cumulative effect of the parameter gradients along the integration path but also effectively bypasses saturation regions, where gradient signals typically vanish. By considering the entire path, this method ensures a more accurate estimation of parameter importance, particularly in regions where simple gradient-based methods may fail due to vanishing gradients or saturation.

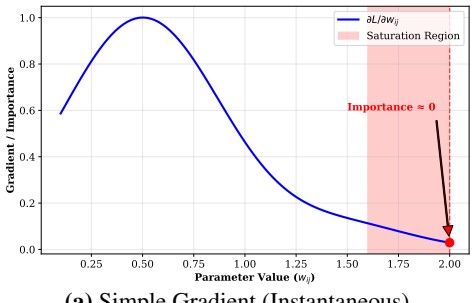 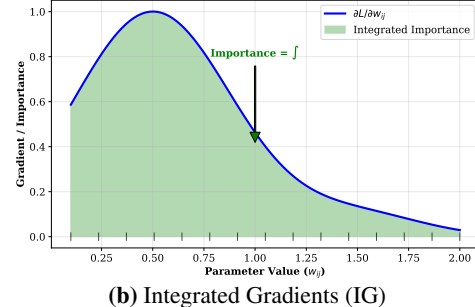

**(a)** Simple Gradient (Instantaneous)      **(b)** Integrated Gradients (IG)

**Figure 2:** Comparison of parameter importance scoring methods. (a) The simple gradient method fails in saturated regions, assigning near-zero importance. (b) Integrated gradients compute importance by integrating along the path from initial to final parameter values, capturing the actual total contribution.

## 3.2 IMPORTANCE SCORING VIA INTEGRATED GRADIENTS

Integrated Gradients (IG (Sundararajan et al., 2017)) is an attribution method originally developed to improve the interpretability of deep neural networks by attributing a model's output to its input features. It quantifies the contribution of each input feature by integrating the gradients of the output with respect to the input, along a path from a baseline to the actual input.

Inspired by this idea, we propose IGU-LoRA, which extends IG to the parameter space for importance estimation in LLMs. Specifically, we integrate the gradients of the loss function with respect to

model parameters along a continuous path from a baseline (e.g., zero) to the actual trained weights, thereby capturing the cumulative influence of each parameter on the training loss function. This parameter-space IG formulation addresses key limitations of conventional gradient-based importance scores, such as limited structural interpretability, over-reliance on local (instantaneous) sensitivity, and susceptibility to gradient saturation. Consequently, it provides more stable and comprehensive estimates of parameter importance for transformer-based LLMs.

Formally, given a weight matrix $\Delta\mathbf{W}$, we denote by $w_{ij}$ its $(i, j)$-th entry, representing a specific weight. Let $\mathcal{L}$ denote the loss function of the LLMs. Since Integrated Gradients (IG) requires a baseline representing a state of no information, we choose 0 as the value for $\Delta\mathbf{W}^{(0)}$ as the baseline, and compute the importance score of $w_{ij}$ under IG as:

$$s_e(w_{ij}) = \left| (w_{ij} - \Delta w_{ij}^{(0)}) \int_{\alpha=0}^1 \frac{\partial \mathcal{L}(\alpha(\Delta\mathbf{W} - \Delta\mathbf{W}^{(0)}))}{\partial w_{ij}} d\alpha \right| = \left| w_{ij} \int_{\alpha=0}^1 \frac{\partial \mathcal{L}(\alpha\Delta\mathbf{W})}{\partial w_{ij}} d\alpha \right|, \quad (4)$$

where $\Delta w_{ij}^{(0)} \in \Delta\mathbf{W}^{(0)}$. Due to the massive number of parameters in LLMs, the loss function $\mathcal{L}$ exhibits strong non-convexity and highly nonlinear dependencies in the parameter space. As a result, Eq. (4) involves a high-dimensional integral that lacks a closed-form solution. To approximate it, we discretize the path $[0, 1]$ into $N$ equal intervals with nodes $\alpha_k = k/N$ ($k = 1, \ldots, N - 1$) and apply the trapezoidal rule, yielding:

$$\hat{s}_e(w_{ij}) \approx \frac{|w_{ij}|}{2N} \left| \frac{\partial \mathcal{L}(0)}{\partial w_{ij}} + 2 \sum_{k=1}^{N-1} \frac{\partial \mathcal{L}(\alpha_k \Delta\mathbf{W})}{\partial w_{ij}} + \frac{\partial \mathcal{L}(\Delta\mathbf{W})}{\partial w_{ij}} \right|. \quad (5)$$

Note that Eq. (5) requires gradient evaluations at $N + 1$ points, which leads to $O(N)$ forward-backward passes for each weight $w_{ij}$, making it computationally expensive in large models. To mitigate this computational burden, we propose a **stochastic approximation** strategy: during fine-tuning, We randomly sample a single integration point $\alpha_k = k/N$ for each mini-batch from a set $\{1/N, \ldots, (N-1)/N\}$ that follows a uniform distribution. Consequently, for the $p$-th mini-batch, the importance score of $w_{ij}$ is approximated as:

$$\hat{s}_e^p(w_{ij}) \approx \frac{|w_{ij}|}{2N} \left| \frac{\partial \mathcal{L}(0)}{\partial w_{ij}} + 2 \frac{\partial \mathcal{L}(\alpha_k \Delta\mathbf{W})}{\partial w_{ij}} + \frac{\partial \mathcal{L}(\Delta\mathbf{W})}{\partial w_{ij}} \right|. \quad (6)$$

At the end of the $t$-th training epoch (which consists of $M$ mini-batches), we compute the aggregated importance score of $w_{ij}$ as follows:

$$s_{agg}(w_{ij}) = \frac{1}{M} \sum_{p=1}^{M} \hat{s}_e^p(w_{ij}). \quad (7)$$

Theorem 1 bounds the error of our estimator, quantifying the gap between the exact IG score in Eq. (4) and the epoch-level estimator in Eq. (7); the total error is $O(N^{-2})$ (discretization) $+ O(M^{-1/2})$ (sampling).

**Theorem 1.** *Let $s_e(w_{ij})$ be the importance score based on Integrated Gradients (IG) as defined in Eq. (4), and let $s_{agg}(w_{ij})$ be the epoch-level estimator as defined in Eq. (7). Define $g_{ij}(\alpha) = \frac{\partial \mathcal{L}(\alpha\Delta\mathbf{W})}{\partial w_{ij}}, \alpha \in [0, 1]$.*

*We assume the following:*

1. *$g_{ij}$ is twice continuously differentiable on $[0, 1]$, and there exists a constant $C_2 < \infty$ such that*

$$\sup_{\alpha \in [0,1]} \left| g_{ij}''(\alpha) \right| \leq C_2. \quad (8)$$

2. *Let $\alpha_1, \alpha_2, \ldots, \alpha_M$ be i.i.d. samples drawn from the discrete uniform distribution over $\left\{ \frac{1}{N}, \frac{2}{N}, \ldots, \frac{N-1}{N} \right\}$, and let $s_{agg}(w_{ij})$ be defined as in Eq. (7).*

*Then, for any $N, M \geq 1$ and $\delta \in (0, 1)$, with probability at least $1 - \delta$, the following bound holds:*

$$|s_e(w_{ij}) - s_{agg}(w_{ij})| \leq \frac{|w_{ij}|C_2}{12N^2} + c|w_{ij}|B\sqrt{\frac{\log(1/\delta)}{M}}, \quad (9)$$

*where $c > 0$ is an absolute constant, and $B$ is a constant such that $|g_{ij}(\alpha)| \leq B$ for all $\alpha \in \left\{ \frac{1}{N}, \frac{2}{N}, \ldots, \frac{N-1}{N} \right\}$. The proof is provided in Appendix A.1.*

## 3.3 Uncertainty-Aware Scoring

Recent studies (Zhang et al., 2022) demonstrate that stochastic sampling and complex training dynamics result in high variance in importance score estimates via Eq. (7), thereby undermining their reliability. To alleviate this issue, we incorporate two complementary mechanisms: sensitivity smoothing and uncertainty quantification, defined respectively as:

$$\bar{s}_e^{(t)}(w_{ij}) = \beta_1 \bar{s}_e^{(t-1)}(w_{ij}) + (1 - \beta_1)s_{agg}^{(t)}(w_{ij}), \tag{10}$$

$$\bar{U}^{(t)}(w_{ij}) = \beta_2 \bar{U}^{(t-1)}(w_{ij}) + (1 - \beta_2)\left|s_{agg}^{(t)}(w_{ij}) - \bar{s}_e^{(t)}(w_{ij})\right|. \tag{11}$$

We define the final importance score as:

$$s_{snr}^{(t)}(w_{ij}) = \mathsf{SNR}_t = \frac{\bar{s}_e^{(t)}(w_{ij})}{\bar{U}^{(t)}(w_{ij}) + \epsilon}, \tag{12}$$

where the numerator $\bar{s}_e^{(t)}(w_{ij})$ captures the persistent influence of the parameter $w_{ij}$ via exponential moving averaging of gradient-parameter correlations. The denominator $\bar{U}^{(t)}(w_{ij})$ quantifies epistemic uncertainty by measuring deviations from the smoothed sensitivity across mini-batches. $\epsilon$ is a very small number to prevent the denominator in Eq. (12) from being 0. This ratio can be interpreted as a signal-to-noise ratio (SNR), providing a criterion for assessing the importance of parameters. Specifically, a larger smoothed sensitivity $\bar{s}_e^{(t)}(w_{ij})$ indicates that $w_{ij}$ consistently exerts strong influence on the loss function. In contrast, a smaller uncertainty $\bar{U}^{(t)}(w_{ij})$ suggests lower variability, reinforcing the reliability of the signal. A high-probability stability guarantee for the EMA ratio score $\mathsf{SNR}_t$ is presented in Appendix A.2. We summarize the complete workflow of IGU-LoRA in Algorithm 1.

---

**Algorithm 1** IGU-LoRA

---

**Input:** Dataset $\mathcal{D}$; the number of total iterations $T$; a pre-trained parameter matrix $\mathbf{W}_0 \in \mathbb{R}^{d_1 \times d_2}$ of a large language model, number of mini-batches $M$; budget of remaining singular values $b$; randomly initialize trainable low-rank matrices $\mathbf{A} \in \mathbb{R}^{d_1 \times r}$ and $\mathbf{B} \in \mathbb{R}^{r \times d_2}$; hyperparameters $\beta_1, \beta_2$.
1: **for** $t = 1$ to $T$ **do**
2:     **for** $p = 1$ to $M$ **do**
3:         Sample a mini-batch from $\mathcal{D}$ and train $\mathbf{A}$ and $\mathbf{B}$.
4:         Perform SVD on the matrix product $\mathbf{AB}$ to obtain $\mathbf{P\Lambda Q} = \mathrm{SVD}(\mathbf{AB})$, where $\mathbf{\Lambda} = diag\{\lambda_1, \lambda_2, \ldots, \lambda_r\}$.
5:         Compute the $\hat{s}_e^p$ in Eq. (6) for every parameter in $\mathbf{P}, \mathbf{Q}$.
6:     **end for**
7:     Compute the aggregated importance score $s_{agg}$ in Eq. (7) for every parameter in $\mathbf{P}, \mathbf{Q}$.
8:     Compute the $\bar{s}_e^{(t)}$ in Eq. (10) and $\bar{U}^{(t)}$ in Eq. (11) for every parameter in $\mathbf{P}, \mathbf{Q}$.
9:     Update the final importance score $s_{snr}^{(t)}$ in Eq. (12).
10:    Compute the importance score of each singular value $S_i$ in Eq. (3) for $\mathbf{P\Lambda Q}$.
11:    Find the top $b$ eigen value: $\hat{\lambda}_1, \hat{\lambda}_2, \ldots, \hat{\lambda}_b$ by importance score $S_i$.
12:    Set $\tilde{\mathbf{\Lambda}} \leftarrow diag(\hat{\lambda}_1, \hat{\lambda}_2, \ldots, \hat{\lambda}_b, 0, \ldots, 0)$ .
13:    Update $\mathbf{A} \leftarrow \mathbf{P}_{:,\pi_{1:b}} \tilde{\mathbf{\Lambda}}^{1/2}$, $\mathbf{B} \leftarrow \tilde{\mathbf{\Lambda}}^{1/2} \mathbf{Q}^{\top}_{\pi_{1:b},:}$   $\triangleright$ The subscript $\pi$ denotes the index set obtained by sorting the columns of $\mathbf{P}$ and $\mathbf{Q}$ in descending order; $\pi_{1:k}$ represents the indices of the first $b$ selected columns; $\mathbf{P}_{1:\pi_b}$ represents selecting the first $b$ columns according to the order defined by $\pi$.
14: **end for**
**Output:** $\mathbf{W} = \mathbf{W}_0 + \mathbf{AB}$

---

## 4 Experiments

### 4.1 Experimental Settings

**Computational Resources.** All experiments are implemented in PyTorch and conducted on an NVIDIA L40 GPU (48GB) running Ubuntu 18.04.1.

**Pretrained Backbone Models.** We use RoBERTa-large model (Liu et al., 2019) as the backbone for the GLUE tasks. For the remaining tasks, we adopt Qwen-2.5-0.5B model [1]. We further validate the robustness and generalization of IGU-LoRA via a backbone ablation, fine-tuning larger-parameter

---

[1] https://huggingface.co/Qwen/Qwen2.5-0.5B

backbones (Llama-2-7B (Touvron et al., 2023), Llama-3-8B (Dubey et al., 2024), DeepSeek-R1-Distill-Qwen-2.5-7B [2]) on multiple datasets.

**IGU-LoRA Configuration.** For the BoolQ, ARC, GSM8K, and AQuA tasks, we perform instruction tuning. The initial LoRA rank is set to $r^{(0)} = 32$, and pruned to an average rank of $r^{(1)} = 16$, achieving pruning 50% rank reduction. For the GLUE tasks, we follow AdaLoRA's setup, using a classification or regression head, with $r^{(0)} = 2$ pruned to an average $r^{(1)} = 1$. During the fine-tuning, IGU-LoRA selects the scaling factor $\alpha$ from $N = 20$ uniformly spaced values in the interval $(0, 1)$. Rank pruning begins at epoch 2 and ends at epoch 5, performed at every one-fifth of an epoch. After pruning, we fine-tune the modules with early stopping (patience = 10 steps) to restore performance. Inference is performed using beam search with a width of 3.

**Reproducibility.** Each task is run with 5 different random seeds, and we report the median test performance. All predictions are generated using the model's language modeling head, which is conditioned on a given prompt or instruction. Additional training configurations are available in Appendix C.

## 4.2 DATASETS AND EVALUATION METRICS

We group the tasks into 2 categories and compare the proposed IGU-LoRA against several baselines: (i) **GLUE Benchmark Datasets** (Wang et al., 2018) include a diverse set of language understanding tasks, such as paraphrase detection (MRPC, QQP), sentiment classification (SST-2), natural language inference (MNLI, RTE, QNLI), and linguistic acceptability (CoLA). (ii) **Mathematical and Common-Sense Reasoning Datasets** include two mathematical reasoning tasks: AQuA (Li et al., 2024) and GSM8K (Cobbe et al., 2021), and four common-sense question answering tasks: ARC-e, ARC-c (Clark et al., 2018), BoolQ (Clark et al., 2019) and COPA (Roemmele et al., 2011). Detailed dataset descriptions, statistical, and evaluation metrics are in Appendix I.

## 4.3 BASELINE METHODS

To evaluate the performance of the proposed IGU-LoRA method in fine-tuning LLMs, we compare it against the following representative baseline: (i) **LoRA and Its Variants.** We evaluate four LoRA-based approaches: LoRA (Hu et al., 2022a), AdaLoRA (Zhang et al., 2023), DoRA (Liu et al., 2024b), AutoLoRA (Xu et al., 2023) and GoRA (He et al., 2025). (ii) **Other PEFT Method.** We also evaluate the following non-LoRA parameter-efficient fine-tuning methods: Housbly-Adapter (Houlsby et al., 2019), P-Tuning v2 (Liu et al., 2021), $(IA)^3$ (Liu et al., 2022a), and SSP (Hu et al., 2022b). (iii) **Full Fine-tuning Method.** For reference, we also include results from full-parameter fine-tuning (denoted as Full FT). All baseline methods are implemented using publicly available codebases. Hyperparameter settings are listed in Appendix C, and additional descriptions of baselines are provided in Appendix J.

## 4.4 MAIN RESULTS

**Table 1:** Performance comparison of fine-tuning methods on the GLUE task using RoBERTa-large. All results are reported as the median over 5 runs with different random seeds. Bold and Underline indicate the best and the second-best results. The metric for each task is explained in Appendix I.5.

| Method | # Params | CoLA (mcc) | SST-2 (acc) | MRPC (acc-f1) | QQP (acc-f1) | STS-B (corr) | MNLI (acc) | QNLI (acc) | RTE (acc) | Avg. |
|---|---|---|---|---|---|---|---|---|---|---|
| Full FT | 355M | 69.19 | 95.63 | 89.46 | **91.10** | 91.60 | 90.01 | 94.03 | 86.94 | 88.50 |
| Housbly-Adapter | 0.35M | 67.80 | 94.38 | 89.75 | 89.41 | 91.08 | 90.28 | 93.52 | 84.36 | 87.57 |
| P-tuning v2 | 0.31M | 67.35 | 93.13 | 88.49 | 88.63 | 90.41 | 89.19 | 91.94 | 82.42 | 86.45 |
| $(IA)^3$ | 0.33M | 68.62 | 93.82 | 89.54 | 89.78 | 90.84 | 89.87 | 92.60 | 82.75 | 87.23 |
| SSP | 0.36M | 69.89 | 94.96 | 90.08 | 90.14 | 91.37 | 90.42 | 94.16 | 84.88 | 88.24 |
| LoRA | 0.33M | 68.71 | 94.84 | 89.71 | 90.26 | 91.63 | 90.34 | 93.87 | 85.56 | 88.12 |
| AdaLoRA | 0.35M | 70.04 | 95.62 | 90.34 | 90.37 | 91.57 | 90.18 | 94.29 | 87.06 | 88.68 |
| DoRA | 0.33M | 70.26 | 95.80 | 90.12 | 90.16 | 91.68 | 90.43 | 94.17 | 87.38 | 88.75 |
| AutoLoRA | 0.34M | 70.47 | 95.53 | 90.26 | 90.31 | 91.52 | 90.26 | 94.08 | 87.64 | 88.76 |
| IGU-LoRA | 0.33M | **71.93** | **96.17** | **90.69** | 90.68 | **91.95** | **90.76** | **94.72** | **88.46** | **89.42** |

**GLUE Benchmark Results.** We evaluate the performance of IGU-LoRA against baseline methods on the GLUE development set using the RoBERTa-large model. The results are presented in Table 1.

---

[2]https://huggingface.co/deepseek-ai/DeepSeek-R1-Distill-Qwen-7B

Under the constraint of fine-tuning only 1% of model parameters, IGU-LoRA achieves performance that is comparable to or surpasses existing approaches across all tasks. Notably, on the CoLA task, IGU-LoRA achieves a Matthews correlation coefficient (MCC) of 71.93%, outperforming the best baseline by 1.5%. On the RTE task, it exceeds the second-best method, AutoLoRA, by 0.8% in accuracy (acc). Similar improvements are also observed on the remaining tasks, demonstrating the robustness of IGU-LoRA. Averaged across all tasks, IGU-LoRA achieves the highest overall performance. Importantly, it maintains strong parameter efficiency, requiring only 0.33 million trainable parameters, comparable to leading PEFT methods, while significantly outperforming full-parameter fine-tuning in both accuracy and efficiency.

**Table 2:** Performance comparison of fine-tuning methods on the Mathematical and common-sense reasoning task using the Qwen-2.5-0.5B. All results are reported as the median over 5 runs with different random seeds. Bold and Underline indicate the best and the second-best results.

| Method | # Params | BoolQ (acc) | ARC-e (acc) | ARC-c (acc) | GSM8K (acc) | AQuA (acc) | Avg. |
|---|---|---|---|---|---|---|---|
| Full FT | 494.0M | 81.74 | **74.82** | 54.98 | **34.64** | 48.72 | 58.98 |
| Housbly-Adapter | 9.0M | 78.36 | 71.04 | 53.26 | 28.67 | 42.85 | 54.84 |
| LoRA | 8.8M | 78.94 | 72.78 | 54.38 | 31.42 | 45.33 | 56.57 |
| AdaLoRA | 8.9M | 80.32 | 73.90 | 54.23 | 33.27 | 46.58 | 57.67 |
| GoRA | 8.8M | 79.24 | 71.20 | 51.91 | 32.07 | 45.81 | 56.04 |
| IGU-LoRA | 8.8M | **82.45** | 74.62 | **55.67** | 34.16 | **48.93** | **59.17** |

**Mathematical and Common-Sense Reasoning Benchmark Results.** We further systematically conduct mathematical and common-sense reasoning tasks using the Qwen-2.5-0.5B model, comparing four representative fine-tuning methods: Full Fine-tuning, Adapter, LoRA, AdaLoRA and GoRA. Table 2 summarizes the results, where IGU-LoRA consistently achieves performance advantages across most tasks. Specifically, IGU-LoRA achieves state-of-the-art results on BoolQ, ARC-c, and AQuA, outperforming the second-best method by 0.2% to 0.8% in accuracy. While it does not obtain the highest score on ARC-e and GSM8K, IGU-LoRA fine-tunes only 8.8M parameters, substantially fewer than full-parameter tuning (494.0M), yet delivering comparable performance. Across all evaluated datasets, IGU-LoRA consistently outperforms other parameter-efficient methods with similar parameter budgets, highlighting its strong generalization under tight resource constraints.

## 4.5 Ablation Study and Analysis

**Analysis of Training and Inference Efficiency.** So far, we have shown that IGU-LoRA outperforms LoRA, AdaLoRA, and DoRA on BoolQ. A natural concern is whether these gains come at the expense of extra time or memory cost. We fine-tune the Qwen-2.5-0.5B model and report peak training GPU memory and wall-clock training time, as well as inference peak GPU memory and decoding latency, as shown in Table 3. All methods utilise a similar memory due to the frozen backbone. LoRA trains the fastest but yields smaller gains; DoRA is slower because it maintains normalized weight directions while updating an additional magnitude vector $\rho$, which involves adding normalization/rescaling operations and optimizer states each step. AdaLoRA improves accuracy via sensitivity-based rank pruning in a two-stage schedule; IGU-LoRA adopts a similar two-stage design and thus achieves comparable training time while delivering higher accuracy. For inference, IGU-LoRA matches LoRA, DoRA, and AdaLoRA in memory usage and decoding latency.

**Table 3:** The time cost, memory and speed for fine-tuning Qwen-2.5-0.5B on the BoolQ task with different PEFT methods.

| Method | Training | | Inference | |
|---|---|---|---|---|
| | Time cost (h) | GPU Mem (GB) | Speed (it/s) | GPU Mem (GB) |
| LoRA | 0.42 | 10.21 | 5.50 | 10.3 |
| AdaLoRA | 0.73 | 10.60 | 5.21 | 10.4 |
| DoRA | 0.95 | 9.53 | 5.30 | 10.3 |
| IGU-LoRA | 0.87 | 10.32 | 5.23 | 10.3 |

**Table 4:** Comparison of the performance of different variants of IGU-LoRA on fine-tuning Qwen-2.5-0.5B across BoolQ and GSM8K tasks.

| Method | BoolQ | GSM8K | Avg. |
|---|---|---|---|
| IGU-LoRA-1 (w/o $\alpha$) | 81.87 | 33.76 | 57.82 |
| IGU-LoRA-2 ($N{=}10$) | 82.14 | 33.95 | 58.05 |
| IGU-LoRA-3 ($N{=}4$) | 82.02 | 33.83 | 57.93 |
| IGU-LoRA-4 ($s_e = \bar{s}_e \cdot \bar{U}$) | 82.28 | 33.69 | 57.99 |
| IGU-LoRA | **82.45** | **34.16** | **58.31** |

**Ablation Study on Hyperparameters and Importance Scoring.** To assess the sensitivity of IGU-LoRA to its key hyperparameters and scoring components, we perform the ablation study by incrementally disabling or simplifying individual modules. Specifically, we evaluate the following variants: (1) IGU-LoRA-1 removes the gradient-integrated $\alpha$ coefficient used during both training and pruning; (2) IGU-LoRA-2 reduces candidate resolution of $\alpha$ from $N = 20$ to $N = 10$; (3) IGU-LoRA-3 further reduces the candidate set to $N = 4$; and (4) IGU-LoRA-4 replaces the final

importance score in Eq. (12) with the alternative formulation in Eq. (11) from Zhang et al. (2023), which combines sensitivity and uncertainty via AdaLoRA's multiplicative strategy [3]. As shown in Table 4, all variants exhibit performance degradation, particularly IGU-LoRA-3 and IGU-LoRA-4, which involve more aggressive simplifications. These results confirm that the default configuration of IGU-LoRA, with high-resolution integrated gradient and uncertainty-aware scoring, is critical in achieving strong performance.

**Hyperparameter Sensitivity Analysis.** To investigate the sensitivity of IGU-LoRA to key hyperparameters, we varied one hyperparameter at a time while keeping others fixed. We analyzed the effects of mini-batch size $M$, the number of discrete points for $\alpha$ (denoted as $N$), and smoothing coefficients $\beta_1$ and $\beta_2$. Experiments were conducted by fine-tuning the Qwen2.5-0.5B model on the Boolq and GSM8K datasets. The results, shown in Figure 3 and 4, demonstrate that IGU-LoRA performs stably across a range of values. Performance improves with larger $M$ and $N$, suggesting better adaptability with finer granularity in scaling factor selection. The coefficients $\beta_1$ and $\beta_2$ show good robustness, with optimal performance in a moderate range. These findings indicate that $M$, $N$, $\beta_1$, and $\beta_2$ are robust hyperparameters for IGU-LoRA.

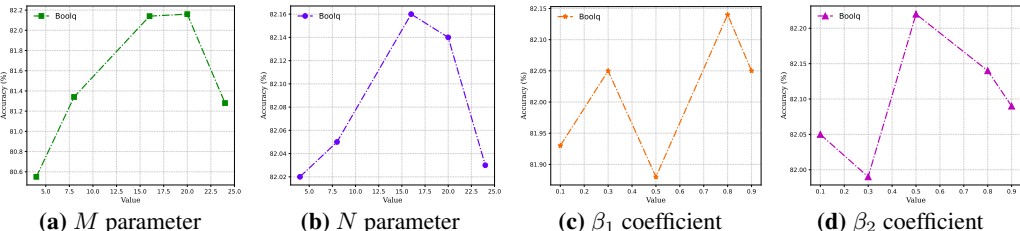

| **(a)** $M$ parameter | **(b)** $N$ parameter | **(c)** $\beta_1$ coefficient | **(d)** $\beta_2$ coefficient |

**Figure 3:** The impact of different hyperparameters $M, N, \beta_1, \beta_2$ on performance when fine-tuning the Qwen2.5-0.5B model on the Boolq dataset. Please zoom in 300% for better clarity.

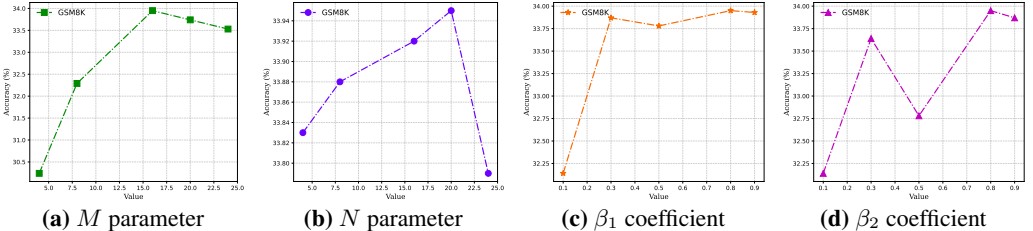

| **(a)** $M$ parameter | **(b)** $N$ parameter | **(c)** $\beta_1$ coefficient | **(d)** $\beta_2$ coefficient |

**Figure 4:** The impact of different hyperparameters $M, N, \beta_1, \beta_2$ on performance when fine-tuning the Qwen2.5-0.5B model on the GSM8K dataset. Please zoom in 300% for better clarity.

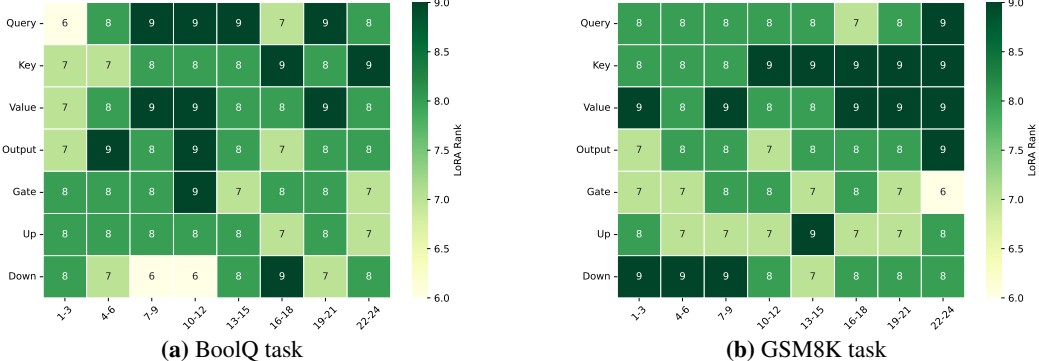

| **(a)** BoolQ task | **(b)** GSM8K task |

**Figure 5:** Rank allocation by IGU-LoRA on the Qwen-2.5-0.5B backbone after fine-tuning for the BoolQ and GSM8K tasks. Please zoom in 300% for better clarity.

**Visualization of Rank Allocation in IGU-LoRA.** Figure 5 visualizes the pruned LoRA rank allocation produced in IGU-LoRA. The rank distributions vary significantly across tasks, underscoring the need for task-specific adaptation to achieve optimal performance. Even within a single task, different Transformer layers allocate ranks differently, reflecting the fine-grained sensitivity of model

---

[3]AdaLoRA (Zhang et al., 2023) for details.

components to low-rank updates. Despite this heterogeneity, consistent structural patterns emerge: in the self-attention mechanism, the Query and Key projections are most frequently prioritized for adaptation, while in the feed-forward network (FFN), the Up and Down projection layers receive the highest ranks. These observations reveal structural preferences in LoRA-based fine-tuning, offering valuable insights for designing generalized and efficient low-rank adaptation strategies.

**Comparisons on Rank Budgets.** In the main experiments, we fixed the initial rank budget at $r^0 = 32$ as a standard configuration. To further evaluate the robustness and adaptability of IGU-LoRA, we vary the initial rank budget across $\{2, 4, 8, 16, 32, 64\}$ and compare its performance with AdaLoRA, LoRA, and DoRA on the BoolQ and GSM8K tasks. The results, shown in Figure 6, demonstrate that IGU-LoRA consistently outperforms AdaLoRA, LoRA and DoRA under all budget settings. This is attributed to its ability to allocate LoRA dynamically across Transformer layers, which enables more effective adaptation.

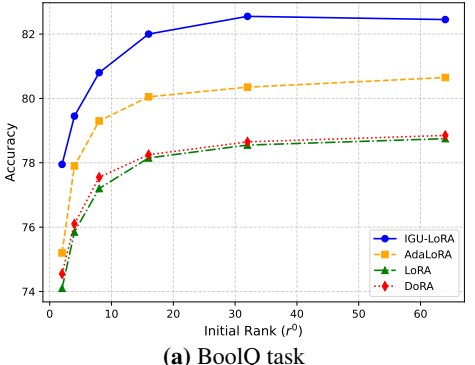 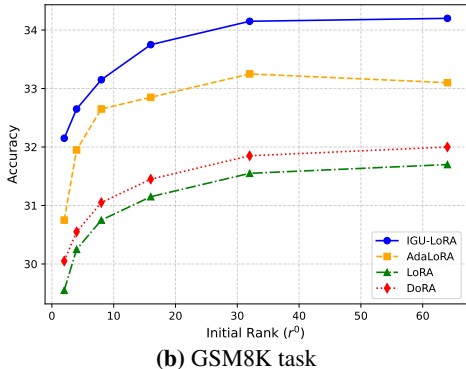

**(a)** BoolQ task      **(b)** GSM8K task

**Figure 6:** Performances across different initial rank budgets. The $x$-axis denotes the initial rank $r^0$, while the $y$-axis indicates the corresponding task performance. Please zoom in 300% for better clarity.

**Comparisons on Different Backbone Models.** To demonstrate the broad applicability of our method, we now conduct experiments on Llama-2-7B, Llama-3-8B and DeepSeek-R1-Distill-Qwen-2.5-7B. The results are reported in Table 5. We can see that on these three backbones, IGU-LoRA can also outperform the baseline methods.

**Table 5:** PEFT methods comparison on different backbones. Left: GLUE accuracy (%) with Llama-2-7B. Right: BoolQ and GSM8K accuracy (%) with Llama-3-8B and DeepSeek-R1-Distill-Qwen-2.5-7B. Results are reported as the median over 5 random seeds. Bold and underline indicate the best and the second-best results.

| Llama-2-7B | | | | | | | |
|---|---|---|---|---|---|---|---|
| Method | # Params | SST-2 | RTE | QNLI | BoolQ | COPA | Avg. |
| Full FT | 6738M | **95.83** | **92.11** | 92.54 | 87.30 | **93.01** | **92.16** |
| Adapter | 21.2M | 94.15 | 82.12 | 93.10 | 87.03 | 91.10 | 89.50 |
| P-tuning v2 | 20.9M | 93.42 | 79.62 | 92.64 | 84.73 | 90.30 | 88.14 |
| SSP | 40.0M | 94.14 | 83.11 | 93.10 | 87.11 | 91.65 | 89.82 |
| LoRA | 20.0M | 94.12 | 83.37 | 93.10 | 87.34 | 91.33 | 89.85 |
| AdaLoRA | 20.0M | 94.12 | 83.51 | 93.20 | 87.11 | 91.62 | 89.91 |
| DoRA | 40.0M | 94.24 | 84.12 | 91.23 | 85.51 | 90.01 | 89.02 |
| IGU-LoRA | 40.0M | 94.34 | 84.33 | **93.33** | **88.11** | 92.10 | 90.44 |

| Llama-3-8B | | | |
|---|---|---|---|
| Method | BoolQ | GSM8K | Avg. |
| LoRA | 88.48 | 73.54 | 81.01 |
| AdaLoRA | 91.65 | 75.82 | 83.74 |
| DoRA | 88.07 | 74.75 | 81.41 |
| IGU-LoRA | **93.33** | **77.63** | **85.48** |
| **DeepSeek-R1-Distill-Qwen-2.5-7B** | | | |
| Method | BoolQ | GSM8K | Avg. |
| LoRA | 88.38 | **74.60** | 81.49 |
| AdaLoRA | 90.54 | 73.30 | 81.92 |
| DoRA | 88.48 | 69.52 | 79.00 |
| IGU-LoRA | **92.82** | 74.28 | **83.55** |

## 5 CONCLUSION

In this work, we address the challenge of parameter importance estimation for efficient fine-tuning of LLMs. We propose IGU-LoRA, a robust scoring framework that integrates the concept of integrated gradients with an uncertainty-aware quantification mechanism. Unlike prior methods that rely solely on instantaneous gradient signals, IGU-LoRA captures each parameter's global and long-term contribution to model performance. Experimental results across diverse tasks and model architectures demonstrate that IGU-LoRA consistently outperforms state-of-the-art baselines, validating its effectiveness and generality. Nevertheless, the method incurs non-trivial computational overhead in network models in networks with large parameter counts, and its performance can be influenced by the choice of integration paths and the precision of uncertainty estimation. In future work, we plan to extend IGU-LoRA to larger-scale models and cross-modal tasks to further explore its adaptability and generalization across architectures.

## 6 ETHICS STATEMENT

This paper proposes an efficient fine-tuning framework, IGU-LoRA, that adaptively allocates LoRA ranks to alleviate the inaccuracy of gradient-sensitivity-based parameter importance estimation under gradient saturation, thereby enhancing the adaptability of large language models (LLMs) across diverse task domains. This study strictly adheres to ethical guidelines: no human subjects or sensitive data were involved. All experimental data are publicly available fine-tuning datasets, and no scenarios containing harmful content were used. While IGU-LoRA effectively improves the overall performance of LLMs, the models may still produce erroneous outputs or misjudgments; thus, we do not recommend deploying them in high-risk scenarios without thorough validation. We further declare that this work has no conflicts of interest, and all experiments and data processing comply with relevant ethical standards.

## 7 REPRODUCIBILITY STATEMENT

For clarity and reproducibility, we summarize the critical details of our method in the main text and Appendix as follows.

- **Algorithmic Details:** We provide a detailed description of the IGU-LoRA algorithm in Section 3, including the integrated gradients computation (Section 3.2) and uncertainty-aware scoring mechanism (Section 3.3). Pseudocode is provided in Algorithm 1.
- **Theoretical Analysis:** We present a theoretical analysis of the approximation error for parameter-space integrated gradients Section 3.2, Appendix A.1 and Appendix A.2, including all necessary assumptions and proofs.
- **Experimental Setup:** We detail the experimental setup in Section 4.1 and Appendix C.
- **Code Availability:** We adopt the code proposed by Zheng et al. (2024) for model training, which is publicly available at `https://github.com/hiyouga/LLaMA-Factory`. In addition, if this work is accepted, we commit to releasing the source code of our method.

## 8 ACKNOWLEDGEMENTS

This work was supported by the Natural Science Foundation of ChongqingJoint Fund for Innovation and Development (Grant No. CSTB2025NSCQ-LZX0134); the Science and Research Development Center of the Ministryof Education (Grant No. 2023ZY024); the Graduate Innovation Project ofChongqing Technology and Business University (Grant No. yjscxx2025-269-181); the Chongqing Technology and Business University ResearchFunds (413/1956019, 413/1952037 and 1752004).

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

# A THEORETICAL PROOFS

## A.1 PROOF OF THEOREM 1

*Proof.* Fix $w_{ij}$ and set $f(\alpha) \equiv f_{ij}(\alpha) = g_{ij}(\alpha) = \partial \mathcal{L}(\alpha \Delta \mathbf{W})/\partial w_{ij}$. By Eq. (4), $s_e(w_{ij}) = |w_{ij}| \left| \int_0^1 f(\alpha) \, d\alpha \right|$. Define the composite trapezoidal approximation and its sampled variant:

$$\mathcal{T}_N = \frac{1}{2N}\left[ f(0) + 2\sum_{k=1}^{N-1} f\left(\tfrac{k}{N}\right) + f(1) \right], \qquad \widetilde{\mathcal{T}}_M = \frac{1}{2N}\left[ f(0) + 2(N-1)\overline{f}_M + f(1) \right], \quad (13)$$

where $\overline{f}_M = \frac{1}{M}\sum_{p=1}^{M} f(\alpha_p)$ with $\alpha_p$ i.i.d. drawn from the discrete uniform distribution on $\{1/N, \ldots, (N-1)/N\}$.

Since $s_{agg}(w_{ij}) = |w_{ij}| |\widetilde{\mathcal{T}}_M|$ and $||x| - |y|| \le |x - y|$, the triangle inequality yields

$$\left| s_e(w_{ij}) - s_{agg}(w_{ij}) \right| \le |w_{ij}| \left| \int_0^1 f - \widetilde{\mathcal{T}}_M \right| \le |w_{ij}| \left( \left| \int_0^1 f - \mathcal{T}_N \right| + |\mathcal{T}_N - \widetilde{\mathcal{T}}_M| \right). \quad (14)$$

**Step 1: discretization error.** By assumption, $f$ is twice continuously differentiable on $[0, 1]$ and $\sup_{\alpha \in [0,1]} |f''(\alpha)| \le C_2$. The standard error bound for the composite trapezoidal rule on $[0, 1]$ (see, e.g., classical numerical analysis texts) yields

$$\left| \int_0^1 f(\alpha) \, d\alpha - \mathcal{T}_N \right| \le \frac{C_2}{12\,N^2}. \quad (15)$$

**Step 2: sampling error.** Let $\mu = \frac{1}{N-1}\sum_{k=1}^{N-1} f\left(\tfrac{k}{N}\right)$ denote the average of $f$ over the $(N-1)$ interior nodes. A simple algebraic manipulation gives

$$|\mathcal{T}_N - \widetilde{\mathcal{T}}_M| = \frac{1}{N}\left| \sum_{k=1}^{N-1} f\left(\tfrac{k}{N}\right) - (N-1)\overline{f}_M \right| = \frac{N-1}{N}|\mu - \overline{f}_M| \le |\mu - \overline{f}_M|. \quad (16)$$

By assumption, $f(\alpha)$ is uniformly bounded on the discretization nodes, which is discussed in detail in Appendix B.1: there exists $B < \infty$ such that $|f(\alpha)| \le B$ for all $\alpha \in \{1/N, \ldots, (N-1)/N\}$. Therefore, each sample $f(\alpha_p)$ lies in $[-B, B]$, and Hoeffding's inequality for bounded random variables implies that, for any $\delta \in (0, 1)$,

$$\Pr\left(|\mu - \overline{f}_M| \ge t\right) \le 2\exp\left(-\frac{2Mt^2}{(2B)^2}\right) = 2\exp\left(-\frac{Mt^2}{2B^2}\right). \quad (17)$$

Setting the right-hand side equal to $\delta$ and solving for $t$ yields that, with probability at least $1 - \delta$,

$$|\mu - \overline{f}_M| \le B\sqrt{\frac{2\log(2/\delta)}{M}} \le cB\sqrt{\frac{\log(1/\delta)}{M}} \quad (18)$$

for an absolute constant $c > 0$. Combining with the previous display gives

$$|\mathcal{T}_N - \widetilde{\mathcal{T}}_M| \le |\mu - \overline{f}_M| \le cB\sqrt{\frac{\log(1/\delta)}{M}} \quad (19)$$

with probability at least $1 - \delta$.

**Step 3: combining the bounds.** Plugging Eq. (15) and Eq. (19) into the decomposition in Eq. (14) yields that, with probability at least $1 - \delta$,

$$\left| s_e(w_{ij}) - s_{agg}(w_{ij}) \right| \le |w_{ij}|\left( \frac{C_2}{12\,N^2} + cB\sqrt{\frac{\log(1/\delta)}{M}} \right), \quad (20)$$

which is exactly the claimed bound in Eq. (9). $\qquad \square$

## A.2 HIGH-PROBABILITY STABILITY OF $\mathsf{SNR}_t$

The resulting SNR-based score favors parameters with consistent, high-impact contributions and suppresses those with volatile or transient behavior. While the above formulation provides an intuitive interpretation of SNR, it remains essential to ensure its statistical stability with high probability, which is formally addressed in Theorem 2.

**Theorem 2.** *Let $y_t = s_{agg}(w_{ij})$ be the per-epoch raw importance defined in Eq. (7). Since $\epsilon$ in Eq. (12) is a very small constant, it can be ignored. Therefore, we have:*

$$\mathsf{SNR}_t = \frac{\bar{s}_e^{(t)}}{\bar{U}^{(t)} + \epsilon} \approx \frac{\bar{s}_e^{(t)}}{\bar{U}^{(t)}}, \tag{21}$$

*Assume that $(y_t)$ is an i.i.d. sequence of sub-Gaussian random variables with mean $\mu$ and variance $\sigma^2$, and let $d = \mathbb{E}\big[|y_t - \mu|\big] > 0$. For $\beta_1, \beta_2 \in (0, 1)$, define the effective EMA window lengths*

$$n_{\mathrm{eff}}(\beta_1) = \frac{1 + \beta_1}{1 - \beta_1}, \quad n_{\mathrm{eff}}(\beta_2) = \frac{1 + \beta_2}{1 - \beta_2}, \quad n_{\mathrm{eff}} = \min\{n_{\mathrm{eff}}(\beta_1), n_{\mathrm{eff}}(\beta_2)\}. \tag{22}$$

*Then there exist universal constants $c_1, c_2, c_0 > 0$ such that, for any $\delta \in (0, 1)$ and all*

$$t \geq t_{\mathrm{burn}} = \left\lceil \frac{c_1}{1 - \min\{\beta_1, \beta_2\}} \log \frac{c_2}{\delta} \right\rceil, \tag{23}$$

*the following holds with probability at least $1 - \delta$:*

$$\big|\mathsf{SNR}_t - \mu/d\big| \leq C \sqrt{\frac{\log(2/\delta)}{n_{\mathrm{eff}}}}, \qquad C = \frac{2\sqrt{2}\,\sigma}{d} + 2c_0 \frac{\mu}{d^2}(\sigma + d). \tag{24}$$

*Proof.* We analyze the EMA under the stylized assumption stated in Theorem 2: $(y_t)$ is an i.i.d. sub-Gaussian sequence with mean $\mu$, variance proxy $\sigma^2$, and $d = \mathbb{E}|y_t - \mu| > 0$.

Recall that Eq. (10) and Eq. (11) define the EMAs

$$\bar{s}_e^{(t)} = \beta_1 \bar{s}_{t-1} + (1 - \beta_1)y_t, \qquad \bar{U}^{(t)} = \beta_2 \bar{U}_{t-1} + (1 - \beta_2)\big|y_t - \bar{s}_e^{(t)}\big|. \tag{25}$$

Unrolling the recursions (for $t$ large enough so that transients are negligible) shows that

$$\bar{s}_e^{(t)} = \sum_{k \geq 0} w_k^{(1)} y_{t-k}, \quad w_k^{(1)} = (1 - \beta_1)\beta_1^k, \qquad \bar{U}^{(t)} = (1 - \beta_2) \sum_{k \geq 0} \beta_2^k \big|y_{t-k} - \bar{s}_{t-k}\big|. \tag{26}$$

Note that $(w_k^{(1)})_{k \geq 0}$ is a geometric weight sequence with $\sum_k w_k^{(1)} = 1$ and

$$\|w^{(1)}\|_2^2 = \sum_{k \geq 0} (1 - \beta_1)^2 \beta_1^{2k} = \frac{1 - \beta_1}{1 + \beta_1} = \frac{1}{n_{\mathrm{eff}}(\beta_1)}. \tag{27}$$

Below we write $n_{\mathrm{eff}} = \min\{n_{\mathrm{eff}}(\beta_1), n_{\mathrm{eff}}(\beta_2)\}$.

**Step 1: concentration of $\bar{s}_e^{(t)}$.** Since $(y_t)$ are i.i.d. sub-Gaussian with mean $\mu$ and variance proxy $\sigma^2$, any fixed weighted sum $\sum_k w_k^{(1)} y_{t-k}$ is also sub-Gaussian with mean $\mu$ and variance proxy $\sigma^2 \|w^{(1)}\|_2^2 = \sigma^2/n_{\mathrm{eff}}(\beta_1)$. Standard sub-Gaussian tail bounds then yield

$$\Pr\left(|\bar{s}_e^{(t)} - \mu| \geq \varepsilon\right) \leq 2\exp\left(-\frac{c\,n_{\mathrm{eff}}(\beta_1)\,\varepsilon^2}{\sigma^2}\right) \tag{28}$$

for an absolute constant $c > 0$. Setting the right-hand side to $\delta/2$ and solving for $\varepsilon$ gives

$$|\bar{s}_e^{(t)} - \mu| \leq \sigma \sqrt{\frac{2\log(4/\delta)}{n_{\mathrm{eff}}(\beta_1)}} \leq \sqrt{2}\,\sigma \sqrt{\frac{\log(4/\delta)}{n_{\mathrm{eff}}}} \tag{29}$$

with probability at least $1 - \delta/2$.

**Step 2: concentration of $\bar{U}^{(t)}$.** We decompose $\bar{U}^{(t)}$ around $d = \mathbb{E}|y_t - \mu|$ as

$$|\bar{U}^{(t)} - d| \leq (1 - \beta_2)\Big|\sum_{k \geq 0} \beta_2^k \big(|y_{t-k} - \mu| - d\big)\Big| + (1 - \beta_2) \sum_{k \geq 0} \beta_2^k \big||y_{t-k} - \bar{s}_{t-k}| - |y_{t-k} - \mu|\big|. \tag{30}$$

Define $X_t = |y_t - \mu| - d$, which is a centered, sub-exponential random variable whose tail parameters depend only on $(\sigma, d)$ (because $y_t$ is sub-Gaussian). Let $w_k^{(2)} = (1 - \beta_2)\beta_2^k$ denote the EMA weights for $\bar{U}^{(t)}$. Then $\sum_{k \geq 0} w_k^{(2)} = 1$ and

$$\|w^{(2)}\|_2^2 = \sum_{k \geq 0} (1 - \beta_2)^2 \beta_2^{2k} = \frac{1 - \beta_2}{1 + \beta_2} = \frac{1}{n_{\text{eff}}(\beta_2)}.$$

Applying a Bernstein-type concentration for weighted sums of i.i.d. sub-exponential variables (see, e.g., standard results on Orlicz norms) yields the existence of an absolute constant $c_0 > 0$ such that, for any $\delta \in (0, 1)$,

$$\Pr\left(\left|(1 - \beta_2)\sum_{k \geq 0} \beta_2^k X_{t-k}\right| \geq c_0(\sigma + d)\sqrt{\frac{\log(4/\delta)}{n_{\text{eff}}(\beta_2)}}\right) \leq \frac{\delta}{2}. \tag{31}$$

For the second term in Eq. (30), note that $\big||a - c| - |a - b|\big| \leq |b - c|$ for any $a, b, c \in \mathbb{R}$, so

$$\big||y_{t-k} - \bar{s}_{t-k}| - |y_{t-k} - \mu|\big| \leq |\bar{s}_{t-k} - \mu|.$$

Thus

$$(1 - \beta_2)\sum_{k \geq 0}\beta_2^k \big||y_{t-k} - \bar{s}_{t-k}| - |y_{t-k} - \mu|\big| \leq (1 - \beta_2)\sum_{k \geq 0}\beta_2^k |\bar{s}_{t-k} - \mu|. \tag{32}$$

We now bound the right-hand side by splitting the sum into a recent window and its tail. Let

$$L = \left\lceil \frac{c_1}{1 - \beta_2}\log\frac{c_2}{\delta}\right\rceil \tag{33}$$

for absolute constants $c_1, c_2 > 0$ chosen large enough. For $t \geq L$, we have

$$(1 - \beta_2)\sum_{k \geq 0}\beta_2^k |\bar{s}_{t-k} - \mu| \leq (1 - \beta_2)\sum_{k=0}^{L}\beta_2^k |\bar{s}_{t-k} - \mu| + (1 - \beta_2)\sum_{k > L}\beta_2^k |\bar{s}_{t-k} - \mu|. \tag{34}$$

For the tail sum, $(1 - \beta_2)\sum_{k > L}\beta_2^k = \beta_2^{L+1}$ and, by choosing $c_1, c_2$ appropriately, we can ensure $\beta_2^{L+1} \leq \delta/(8c_2)$. For the finite window $\{t, t-1, \ldots, t-L\}$, we apply Eq. (29) and a union bound over these $(L + 1)$ indices to obtain, with probability at least $1 - \delta/2$,

$$|\bar{s}_{t-k} - \mu| \leq \sqrt{2}\,\sigma\sqrt{\frac{\log(4L/\delta)}{n_{\text{eff}}(\beta_1)}} \quad \text{for all } 0 \leq k \leq L. \tag{35}$$

Combining these bounds and using $n_{\text{eff}} \leq n_{\text{eff}}(\beta_1)$ yields

$$(1 - \beta_2)\sum_{k \geq 0}\beta_2^k |\bar{s}_{t-k} - \mu| \leq \tilde{c}\,\sigma\sqrt{\frac{\log(2/\delta)}{n_{\text{eff}}}} \tag{36}$$

with probability at least $1 - \delta/2$, for an absolute constant $\tilde{c} > 0$.

Putting Eq. (31) and Eq. (36) back into Eq. (30) and recalling that $n_{\text{eff}} \leq n_{\text{eff}}(\beta_2)$, we obtain that, for $t \geq t_{\text{burn}}$ and with probability at least $1 - \delta$,

$$|\bar{U}^{(t)} - d| \leq C_2'(\sigma + d)\sqrt{\frac{\log(2/\delta)}{n_{\text{eff}}}} \tag{37}$$

for an absolute constant $C_2' > 0$. By increasing $c_1$ if necessary, we may ensure that the right-hand side in Eq. (37) is at most $d/2$, so that $\bar{U}^{(t)} \geq d/2$ holds on the same high-probability event.

**Step 3: bounding the ratio** $\mathsf{SNR}_t$. On the event $\{\bar{U}^{(t)} \geq d/2\}$ we can control the ratio $\mathsf{SNR}_t = \bar{s}_e^{(t)}/\bar{U}^{(t)}$ via the deterministic inequality

$$\left| \frac{\bar{s}_e^{(t)}}{\bar{U}^{(t)}} - \frac{\mu}{d} \right| \leq \frac{2}{d} |\bar{s}_e^{(t)} - \mu| + \frac{2\mu}{d^2} |\bar{U}^{(t)} - d|. \tag{38}$$

Combining Eq. (29) and Eq. (37) with Eq. (38), and noting that $n_{\text{eff}} \leq n_{\text{eff}}(\beta_1)$, gives

$$\left| \mathsf{SNR}_t - \mu/d \right| \leq \left( \frac{2\sqrt{2}\,\sigma}{d} + 2c_0 \frac{\mu}{d^2}(\sigma + d) \right) \sqrt{\frac{\log(2/\delta)}{n_{\text{eff}}}} \tag{39}$$

with probability at least $1 - \delta$, for a suitable absolute constant $c_0 > 0$. This is exactly the claimed bound in Theorem 2 after setting $C = \frac{2\sqrt{2}\,\sigma}{d} + 2c_0 \frac{\mu}{d^2}(\sigma + d)$ and $t_{\text{burn}} = \left\lceil \frac{c_1}{1 - \min\{\beta_1, \beta_2\}} \log \frac{c_2}{\delta} \right\rceil$. $\quad\square$

# B THE DISCUSSION OF THE ASSUMPTIONS IN THEOREM

## B.1 THE ANALYSIS OF THE ASSUMPTION IN THEOREM 1

In this section, we focus on how the assumption in Theorem 1, that $g_{ij}$ is twice continuously differentiable on the interval $[0, 1]$ with a bounded second derivative, leads to the conclusion that $g_{ij}(\alpha)$ is bounded. First, consider the following form of $g_{ij}(\alpha)$:

$$g_{ij}(\alpha) = \frac{\partial \mathcal{L}(\alpha \Delta \mathbf{W})}{\partial w_{ij}}, \quad \alpha \in [0, 1], \tag{40}$$

The analysis of Theorem 1 relies solely on the assumption that $g_{ij}$ is twice differentiable on the interval $[0, 1]$ and that its second derivative is bounded, which allows the application of the composite trapezoidal rule, leading to a discretization error of $O(N^{-2})$. Specifically, numerical analysis typically assumes the existence of a constant $C_2 < \infty$ such that:

$$\sup_{\alpha \in [0, 1]} \left| g_{ij}''(\alpha) \right| \leq C_2. \tag{41}$$

Under this assumption, we can derive the following error bound:

$$\left| \int_0^1 g_{ij}(\alpha)\,d\alpha - \mathcal{T}_N \right| \leq \frac{C_2}{12 N^2}, \tag{42}$$

This equation provides the theoretical basis for the $O(N^{-2})$ discretization error term in Theorem 1. This requirement is essentially a standard smoothness assumption in trapezoidal integration and does not involve any specific distributional assumptions. Furthermore, the condition of bounded second derivatives directly implies that $g_{ij}$ itself is bounded. By the fundamental theorem of calculus:

$$g_{ij}'(\alpha) = g_{ij}'(0) + \int_0^\alpha g_{ij}''(t)\,dt, \ g_{ij}(\alpha) = g_{ij}(0) + \int_0^\alpha g_{ij}'(t)\,dt, \tag{43}$$

We can obtain the bound for all $\alpha \in [0, 1]$:

$$|g_{ij}'(\alpha)| \leq |g_{ij}'(0)| + \int_0^1 |g_{ij}''(t)|\,dt \leq |g_{ij}'(0)| + C_2, \tag{44}$$

Thus,

$$|g_{ij}(\alpha)| \leq |g_{ij}(0)| + \int_0^1 |g_{ij}'(t)|\,dt \leq |g_{ij}(0)| + |g_{ij}'(0)| + C_2 \triangleq B. \tag{45}$$

This implies that $g_{ij}(\alpha)$ is bounded on $[0, 1]$. When we sample $\alpha$ from the finite set $\{1/N, \ldots, (N-1)/N\}$, the resulting random variable $g_{ij}(\alpha)$ is bounded by constant $B$.

## B.2 THE ANALYSIS OF THE I.I.D. ASSUMPTION IN THEOREM 2

Theorem 2 assumes that the per-epoch raw scores $y_t = s_{agg}(w_{ij})$ form an i.i.d. sub-Gaussian sequence with a common mean $\mu$ and variance $\sigma^2$. However, strictly speaking, $y_t$ depends on the current model parameters $\mathbf{W}^{(t)}$, which are updated across epochs, so exact i.i.d. is an idealization.

Our goal is to model the regime in which the training dynamics have *stabilized*: after an initial transient phase (discarded via the burn-in time $t_{\text{burn}}$), the statistics of the gradient noise around the current solution change only slowly. Furthermore, within the effective EMA window $n_{\text{eff}}(\beta_1, \beta_2)$, the gradient sequence can be approximated as having nearly stationary mean and variance. In this regime, standard extensions of EMA concentration results to weakly dependent or mixing sequences apply. We chose the i.i.d. setting for clarity of presentation and to keep the notation simple. It is important to note that Theorem 2 is derived under this stylized, locally stationary noise assumption, and is meant to provide intuition about how the EMA window size and variance control the stability of $\mathsf{SNR}_t$, rather than to capture every aspect of LLM training dynamics exactly.

To support this approximation empirically, we provide a small diagnostic in Appendix G: for a representative layer on BoolQ, we plot the time series of $y_t$ and its running mean/variance across epochs. We observe that, after the early epochs, both the mean and variance of $y_t$ quickly settle into a narrow band, and the lag-1 autocorrelation becomes small. Correspondingly, the $\mathsf{SNR}_t$ curves are nearly flat after burn-in. These observations suggest that, in the regime where EMA-based importance is actually used for rank pruning, the i.i.d./local stationarity approximation is reasonably accurate.

Finally, we emphasize that these assumptions are used only in our theoretical analysis; the algorithm itself does not rely on them. Even when the exact assumptions are relaxed, the qualitative conclusions remain the same: (i) our IG estimator trades off discretization error $O(N^{-2})$ and sampling error $O(M^{-1/2})$, and (ii) EMA-based $\mathsf{SNR}_t$ scores become more stable as the effective sample size increases and the process enters a locally stationary regime.

## C HYPERPARAMETER SETTINGS

During the training process, we tune the learning rate from $\{5 \times 10^{-4}, 1 \times 10^{-4}, 5 \times 10^{-4}, 1 \times 10^{-3}, 2 \times 10^{-4}\}$ and pick the best learning rate for every method. For the MNLI, QNLI, and QQP, we set the batch size to 128. For RTE, MRPC, CoLA, and STS-B, the batch size is set to 32. For SST-2, we use a batch size of 64. For all other tasks, the batch size is set to 16. All baseline methods follow the same settings as IGU-LoRA, as detailed in Table 6. In IGU-LoRA, several key hyperparameters $\epsilon, M, N, \beta_1, \beta_2$ are set to $1 \times 10^{-6}$, 16, 20, 0.85, and 0.85, respectively, as detailed in Table 7. They remain constant throughout the experiment, and their sensitivity is discussed in the main text.

**Table 6:** Hyperparameter setup of IGU-LoRA for training on different datasets.

| Dataset | learning rate | batch size | Max. Sequence Length | # epochs | $\gamma$ | $t_i$ | $\Delta_T$ | $t_f$ |
|---|---|---|---|---|---|---|---|---|
| MNLI | $5 \times 10^{-4}$ | 128 | 512 | 25 | 0.1 | 500 | 20 | 10000 |
| RTE | $1 \times 10^{-3}$ | 32 | 512 | 25 | 0.1 | 300 | 5 | 2500 |
| QNLI | $5 \times 10^{-4}$ | 128 | 512 | 25 | 0.1 | 400 | 20 | 10000 |
| MRPC | $1 \times 10^{-3}$ | 32 | 512 | 25 | 0.1 | 300 | 5 | 2500 |
| QQP | $5 \times 10^{-4}$ | 128 | 512 | 25 | 0.1 | 500 | 20 | 10000 |
| SST-2 | $1 \times 10^{-3}$ | 64 | 512 | 25 | 0.1 | 400 | 20 | 5000 |
| CoLA | $1 \times 10^{-3}$ | 32 | 512 | 25 | 0.1 | 300 | 5 | 2500 |
| STS-B | $2 \times 10^{-3}$ | 32 | 512 | 25 | 0.1 | 300 | 5 | 2500 |
| BoolQ | $5 \times 10^{-4}$ | 16 | 512 | 25 | 0.1 | 500 | 20 | 10000 |
| ARC-e | $5 \times 10^{-4}$ | 16 | 512 | 25 | 0.1 | 500 | 20 | 10000 |
| ARC-c | $5 \times 10^{-4}$ | 16 | 512 | 25 | 0.1 | 500 | 20 | 10000 |
| COPA | $1 \times 10^{-3}$ | 16 | 512 | 25 | 0.1 | 500 | 20 | 10000 |
| AQuA | $1 \times 10^{-4}$ | 16 | 512 | 25 | 0.1 | 500 | 20 | 10000 |
| MMLU | $1 \times 10^{-4}$ | 128 | 512 | 15 | 0.1 | 500 | 20 | 10000 |
| VQA | $2 \times 10^{-4}$ | 32 | 512 | 25 | 0.1 | 300 | 20 | 10000 |
| GAQ | $5 \times 10^{-4}$ | 32 | 512 | 25 | 0.1 | 300 | 20 | 10000 |
| MVLR[2] | $5 \times 10^{-4}$ | 32 | 512 | 25 | 0.1 | 300 | 20 | 10000 |
| COCO | $2 \times 10^{-4}$ | 32 | 512 | 25 | 0.1 | 300 | 20 | 10000 |

**Table 7:** Setting of the 5 hyperparameters ($\epsilon, M, N, \beta_1, \beta_2$) in IGU-LoRA.

| Hyperparameter | $\epsilon$ | $M$ | $N$ | $\beta_1$ | $\beta_2$ |
|---|---|---|---|---|---|
| Value | $1 \times 10^{-6}$ | 16 | 20 | 0.85 | 0.85 |

# D   ABLATION STUDY ON HIGH-IMPACT PARAMETERS

To further validate the effectiveness of IGU-LoRA in identifying high-impact parameters, we conduct an ablation study on high-impact parameters. Specifically, we remove the high-rank and low-rank modules with the highest IGU-LoRA scores from different layers of the Qwen2.5-0.5B model and evaluate the performance drop on the Boolq and GSM8K datasets. As shown in Table 8, removing the high-rank modules from the K module in Layer 3 (L3_K) and the V module in Layer 10 (L10_V) results in a performance drop of 1.30 and 1.33 points on Boolq, respectively. Similarly, removing the high-rank modules from the Q module in Layer 22 (L22_Q) and the K module in Layer 17 (L17_K) results in performance drops of 1.80 and 1.73 points on GSM8K, respectively. In contrast, removing the low-rank modules from the K module in Layer 1 (L1_K) and the V module in Layer 3 (L3_V) results in only minor performance drops of 0.05 and 0.10 points on Boolq, respectively. The same trend is observed on GSM8K when removing the low-rank modules from the Q module in Layer 8 (L8_Q) and the K module in Layer 6 (L6_K), resulting in performance drops of 0.11 and 0.15 points, respectively. These results demonstrate that IGU-LoRA effectively identifies high-impact parameters, as their removal leads to significant performance degradation compared to low-impact parameters.

**Table 8:** Ablation study on the impact of removing high-rank and low-rank modules from different layers on Qwen2.5-0.5B model performance. The numbers in parentheses indicate the performance drop compared to the model with no modules removed. The left table and the right table represent results on Boolq and GSM8K, respectively.

| | Module Removed | Rank | Boolq | | Module Removed | Rank | GSM8K |
|---|---|---|---|---|---|---|---|
| 1 | L3_K | 10 | 81.15 (-1.30) | 1 | L22_Q | 12 | 32.35 (-1.80) |
| 2 | L10_V | 10 | 81.12 (-1.33) | 2 | L17_K | 11 | 32.42 (-1.73) |
| 3 | L3_K / L10_V | 10 / 10 | 80.44 (-2.01) | 3 | L22_Q / L17_K | 12 / 11 | 31.15 (-3.00) |
| 4 | L1_K | 5 | 82.40 (-0.05) | 4 | L8_Q | 6 | 34.05 (-0.11) |
| 5 | L3_V | 5 | 82.35 (-0.10) | 5 | L6_K | 6 | **34.01 (-0.15)** |
| 6 | L1_K / L3_V | 5 / 5 | 82.30 (-0.15) | 6 | L8_Q / L6_K | 6 / 6 | 33.84 (-0.32) |
| 7 | - | - | **82.45** | 7 | - | - | **34.16** |

# E   GENERALIZATION SUPPLEMENTARY EXPERIMENTS

To further validate the generalization performance of IGU-LoRA, we conduct additional experiments on the MMLU benchmark using the Llama2-7B model. As shown in Table 9, IGU-LoRA achieves an average accuracy of $51.07\%$, which is very close to the full fine-tuning method ($51.54\%$) and outperforms LoRA ($49.94\%$). Notably, IGU-LoRA demonstrates superior performance in Science, Technology, Engineering, and Mathematics (STEM) and Social Science subjects, achieving accuracies of $41.71\%$ and $58.12\%$, respectively. These results further confirm the effectiveness of IGU-LoRA in enhancing the generalization capabilities of fine-tuned models across diverse subject areas.

**Table 9:** The generalization performance of fine-tuning the Llama2-7B model on the MMLU benchmark using different methods, reporting the average results over 5 random seeds.

| Method | Humanities | STEM | Social. | Other | Avg. |
|---|---|---|---|---|---|
| Full FT | **49.91** | 41.70 | 57.53 | 57.02 | **51.54** |
| LoRA | 46.15 | 40.84 | 56.63 | 56.23 | 49.94 |
| IGU-LoRA | 47.33 | **41.71** | **58.12** | **57.10** | 51.07 |

# F   MULTIMODAL BENCHMARK SUPPLEMENTARY EXPERIMENTS

To further demonstrate the effectiveness of IGU-LoRA in multimodal tasks, we conduct additional experiments on the VQAv2, GAQ, NVLR[2] and COCO Captioning datasets using the VL-BART (Su et al., 2019). As shown in Table 10, IGU-LoRA achieves an average score of 77.47, outperforming

LoRA (74.31) and DoRA (77.40), and closely approaching the performance of full fine-tuning (77.35). These results further validate the capability of IGU-LoRA to effectively adapt multimodal models while maintaining high performance across different tasks.

**Table 10:** Performance comparison of different fine-tuning methods on the VQA, GAQ, NVLR$^2$ and COCO datasets using the VL-BART model. The results are averaged over 5 random seeds.

| Method | VQAv2 | GAQ | NVLR$^2$ | COCO Captioning | Avg. |
|---|---|---|---|---|---|
| Full FT | 66.91 | 56.72 | 73.71 | 112.04 | 77.35 |
| LoRA | 64.32 | 54.10 | 71.25 | 109.56 | 74.31 |
| DoRA | 65.81 | 54.71 | 73.14 | 115.93 | 77.40 |
| IGU-LoRA | 65.78 | 55.32 | 73.42 | 115.36 | **77.47** |

## G  THE VERIFICATION OF THE I.I.D./LOCAL STATIONARITY APPROXIMATION IN THEOREM 2.

To validate the i.i.d. / local stationarity approximation used in Theorem 2, we conduct an empirical analysis of the importance score statistics during the fine-tuning process. Specifically, we monitor several representative modules (e.g., the L16_Q module for the 16-th layer's Q component and the L5_K module for the 5-th layer's K component) across multiple training iterations on the BoolQ dataset. We observe that, after the initial epochs, the mean and variance of $y_t$ quickly stabilize within a narrow range, and the first-order lag autocorrelation becomes very small. Correspondingly, the $\text{SNR}_t$ curve becomes nearly flat after the burn-in period. These observations suggest that the i.i.d./local stationarity approximation is reasonable and accurate during the stage when EMA-based importance-ranking pruning is applied in practice.

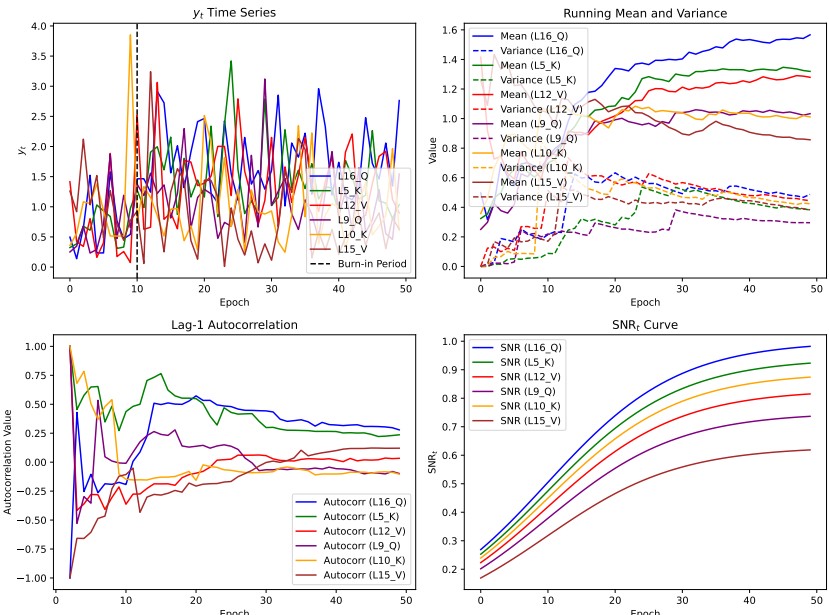

**Figure 7:** Empirical analysis of importance score statistics during fine-tuning. The plots show the changes in $y_t$, the mean and variance of $y_t$, the first-order lag autocorrelation, and $\text{SNR}_t$ across training iterations for representative module parameters.

## H  EFFECTS OF SAMPLE ORDER AND BATCH SIZE

To investigate the effects of sample order and batch size on the performance of IGU-LoRA, we conduct experiments using the Qwen-2.5-0.5B model on the BoolQ dataset. The results are summarized as follows:

**Sample Order / Random Seed.** we trained with a fixed batch size using five different random seeds. These seeds control the data shuffling and the sampled integration nodes $\alpha_k$. The downstream accuracy varies slightly across seeds (within $\Delta_{\text{acc}}$ absolute points, indicating a small change), which demonstrates that the sample order has high stability on the results.

**Batch Size.** We further vary the batch size (e.g., 2, 4, 8, 16, 32) while keeping all other hyperparameters fixed. The resulting test accuracy again shows only minor variation. This proves that batch size does not have a significant impact on the results. The detailed results are presented in Table 11.

**Table 11:** Effect of Batch Size on BoolQ Accuracy across Different Random Seeds

| Batch Size | Seed 1 | Seed 2 | Seed 3 | Seed 4 | Seed 5 |
|:---:|:---:|:---:|:---:|:---:|:---:|
| 2 | 82.46 | 82.47 | 82.45 | 82.46 | 82.45 |
| 4 | 82.45 | 82.46 | 82.44 | 82.45 | 82.44 |
| 8 | 82.44 | 82.45 | 82.43 | 82.44 | 82.43 |
| 16 | 82.45 | 82.46 | 82.44 | 82.45 | 82.44 |
| 32 | 82.40 | 82.41 | 82.39 | 82.40 | 82.39 |

# I DATASETS AND METRICS

## I.1 GLUE BENCHMARK TASKS

**Single-sentence Classification Tasks.** (1) *CoLA (Corpus of Linguistic Acceptability)*: Determine whether a sentence adheres to grammatical rules (binary classification). (2) *SST-2 (Stanford Sentiment Treebank)*: Movie review sentiment analysis (positive/negative binary classification).

**Sentence-pair Classification Tasks.** (1) *MRPC (Microsoft Research Paraphrase Corpus)*: Determine whether two sentences are semantically equivalent (binary classification). (2) *QQP (Quora Question Pairs)*: Determine whether two Quora questions are semantically identical (binary classification). (3) *RTE (Recognizing Textual Entailment)*: Determine whether a sentence pair entails a relationship (three-class classification: entailment/contradiction/neutral).

**Similarity and Regression Task.** *STS-B (Semantic Textual Similarity Benchmark)*: Calculate the semantic similarity between two sentences (continuous value from 1 to 5).

**Question-answering Task**. *QNLI (Question-answering NLI).* Determine whether a sentence contains the answer to a given question (binary classification).

**Natural Language Inference Task.** *MNLI (Multi-Genre Natural Language Inference).* Large-scale cross-domain textual entailment classification (three-class classification).

## I.2 MATHEMATICAL AND COMMON-SENSE REASONING TASKS

**Mathematical Reasoning Tasks.** (1) *AQuA (Algebra question answering)*: Derive the correct answer from a given algebraic problem (multiple-choice) and generate the corresponding solution process (Rationales). (2) *GSM8K (Grade school math 8K)*: Perform multi-step reasoning on mathematical problems described in natural language.

**Common-Sense Reasoning Tasks.** (1) *BoolQ (Boolean questions).* Determine whether the answer to a given question, based on the provided paragraph, is "Yes" (True) or "No" (False). (2) *ARC-e (AI2 reasoning challenge - easy)*: Select the most reasonable answer from a given set of scientific questions (Multiple-choice question). (3) *ARC-c (AI2 reasoning challenge - challenge)*: Combine multi-step reasoning and cross-domain knowledge to provide answers. (4) *COPA (Choice of plausible alternatives).* Select the most plausible cause or effect for a given premise from two provided alternatives. The task requires understanding of causal relationships and commonsense reasoning in everyday scenarios.

## I.3 MULTIMODAL BENCHMARK TASKS

**Visual Question Answering Tasks.** (1) *VQAv2 (Visual Question Answering v2).* Given an image and a related question, select the most appropriate answer from multiple choices. (2) *GAQ (Generalized*

*Question Answering).* This task extends VQA to a more generalized setting, where the model is asked to answer a wider range of questions based on visual context.

**Visual-Linguistic Reasoning Task.** (1) *NLVR2 (Natural Language for Visual Reasoning 2).* Given a pair of images and a natural language statement, determine whether the statement accurately describes the relationship between the two images.

**Image Captioning Task.** (1) *COCO Captioning.* Generate descriptive captions for images in the COCO dataset, evaluating the model's ability to understand and describe visual content accurately.

**Table 12:** Summary of the benchmark datasets.

| Datasets | # train | # dev | # test | Type | Metrics |
|---|---|---|---|---|---|
| *Common-Sense reasoning tasks* | | | | | |
| BoolQ | 9427 | - | 3270 | Common-Sense reasoning | Acc |
| ARC-e | 2251 | 570 | 2376 | Common-Sense reasoning | Acc |
| ARC-c | 1119 | 299 | 1172 | Common-Sense reasoning | Acc |
| COPA | 400 | 100 | 500 | Common-Sense reasoning | Acc |
| *Mathematical reasoning tasks* | | | | | |
| AQuA | 97467 | 254 | 254 | Mathematical reasoning | Acc |
| GSM8K | 7473 | - | 1319 | Mathematical reasoning | Acc |
| *GLUE benchmark tasks* | | | | | |
| SST-2 | 67k | 872 | 1.8k | Sentiment | Acc |
| MNLI | 393k | 20k | 20k | NLU | Acc |
| QQP | 364k | 40k | 391k | Paraphrase | Acc-F1 |
| MRPC | 3.7k | 408 | 107k | Paraphrase | Acc-F1 |
| RTE | 2.5k | 176 | 3k | NLU | Acc |
| QNLI | 108k | 5.7k | 5.7k | QA/NLI | Acc |
| CoLA | 8.5k | 1k | 1k | Acceptability | Mcc |
| STS-B | 7k | 1.5k | 1.4k | Similarity | Corr |

## I.4 DATASET STATISTICS

In our experiments, we compare performance across multiple tasks, including the GLUE benchmark, which consists of eight datasets: CoLA, SST-2, MRPC, QQP, STS-B, MNLI, QNLI, and RTE; three common-sense reasoning tasks (BoolQ, ARC-e, and ARC-c); and two mathematical reasoning tasks (AQuA and GSM8K). The dataset statistics are presented in Table 12.

## I.5 EVALUATION METRICS

As shown in Table 12, we strictly follow the official settings of GLUE and use the same metrics as Wang et al. (2018). For MNLI, we report the average of the accuracy scores on the matched and mismatched test sets. For MRPC and QQP, we report Acc-F1, the average accuracy, and F1 scores. For STS-B, we report Corr, which denotes the average of the Pearson and Spearman correlation coefficients. For CoLA, we report Mcc, which is the Matthews correlation. For all other tasks, we report accuracy (Acc). Since the common sense and math reasoning tasks usually come with a definite answer choice, we will directly consider the correctness of the final answers. Thus, we report accuracy (denoted as Acc).

## J BASELINE DETAILS

• *Full fine-tuning* is the most common approach for adaptation. During fine-tuning, the model is initialized with pre-trained weights and biases, and all model parameters undergo gradient updates.

• *LoRA* (Hu et al., 2022a) is a representative parameter-efficient fine-tuning (PEFT) method. It introduces two low-rank matrices to parameterize the incremental weight updates, and only these lightweight components are updated during fine-tuning. The number of trainable parameters is

determined by the rank $r$ and the number of inserted adaptation matrices $n$, allowing for fine-grained control over the adaptation budget.

• *AdaLoRA* (Zhang et al., 2023) extends the conventional LoRA framework by introducing a dynamic rank adaptation mechanism. It parameterizes the low-rank adapters using singular value decomposition (SVD), and evaluates the importance of each parameter based on the magnitude of its corresponding singular value. This importance score then guides a progressive rank pruning process, allowing the model to dynamically reallocate its limited parameter budget to more critical layers or modules.

• *DoRA* (Liu et al., 2024b) enhances the learning capacity and adaptability of pretrained models by decoupling weight matrices into two distinct components: magnitude and direction. The key idea is to keep the magnitude fixed and apply LoRA-style low-rank updates only to the directional component. This separation allows for more expressive and geometry-aware adaptation while preserving the norm of the original weights, which helps stabilize training and maintain alignment with the pretrained model. Since only the direction is modified, DoRA introduces no additional inference overhead, making it efficient and scalable for deployment.

• *AutoLoRA* (Xu et al., 2023) is a meta-learning-based fine-tuning approach designed to automatically determine the optimal rank for each layer in Low-Rank Adaptation (LoRA). It introduces a learnable selection variable for each rank-1 matrix and dynamically adjusts these variables using a meta-learning strategy. By jointly optimizing the rank configuration along with the LoRA parameters, AutoLoRA significantly improves fine-tuning efficiency and overall performance.

• *Adapter* (Houlsby et al., 2019) inserts lightweight bottleneck modules between each layer of the pretrained model, updating only these newly introduced modules during fine-tuning while keeping the original model parameters frozen.

• *P-tuning v2* (Liu et al., 2021) is an improved prompt tuning method that inserts trainable prompt tokens at the input layer and across multiple model layers. This design increases the trainable parameters from approximately 0.01% to 0.1%-3% of the full model, while maintaining parameter efficiency. P-tuning v2 enhances optimization stability and improves performance across various tasks by integrating task-specific information deeper into the model.

• *(IA)³* (Liu et al., 2022a) introduces learnable scaling vectors at key locations in the Transformer architecture, such as the keys and values in the self-attention mechanism and the intermediate activations in the feed-forward networks. These vectors are applied via element-wise multiplication to modulate the internal activations, enabling flexible control over the model's output without modifying the original model parameters.

• *SSP* (Hu et al., 2022b) leverages structural sparsity to guide the automatic search for parameter insertion locations, activating trainable parameters only in the most important substructures. This enables higher efficiency without sacrificing model performance.

• *GoRA* (He et al., 2025) leverages gradient-driven adaptive low-rank adjustment to dynamically adjust the rank of low-rank adaptation layers during training. By using gradient information, GoRA ensures that the model can allocate computational resources more efficiently, adjusting the rank based on the importance of each layer for different tasks and training stages. This method maintains computational efficiency while improving model performance, adapting the low-rank configuration to meet the specific needs of the training process.

# K  ADDITIONAL RELATED WORKS

## K.1  DYNAMIC RANK ALLOCATION

Dynamic rank allocation gains increasing attention in deep learning model optimization, with various methods proposed to improve adaptability and efficiency. Several other notable approaches are introduced beyond AdaLoRA (Zhang et al., 2023) and AutoLoRA (Xu et al., 2023). LoSA (Huang et al., 2025) integrates sparsity and low-rank adaptation, dynamically adjusting both using representation mutual information and reconstruction error. PRILoRA (Benedek & Wolf, 2024) employs a heuristic strategy that linearly increases ranks from lower to higher layers, motivated by the observation that higher layers often require greater adaptability in transfer learning. ALoRA (Liu et al., 2024c) further incorporates a novel mechanism, AB-LoRA, which assesses the importance of individual LoRA

ranks and incrementally prunes redundant components, reallocating the freed budget to more critical Transformer modules. These methods provide diverse rank allocation strategies that contribute to more efficient fine-tuning of large models.

## L  THE USE OF LARGE LANGUAGE MODELS

During the preparation of this manuscript, large language models (LLMs) were employed in several auxiliary capacities. First, at the writing stage, LLMs were utilized to refine and translate the text, thereby enhancing the overall fluency, readability, and precision of academic expression. Second, in relation to experiments and results presentation, LLMs assisted in generating parts of the code for data visualization and figure plotting, which facilitated a more efficient presentation of research findings. Third, in surveying the research landscape and related work, LLMs provided support for literature searches, helping us to locate and summarize relevant studies in the field systematically. Finally, in the theoretical component of this work, LLMs offered auxiliary support in structuring complex proofs and verifying critical derivation steps, contributing to the clarity and rigor of our theoretical analysis. It should be emphasized that all uses of LLMs were strictly auxiliary in nature; the formulation of research questions, the design of methods, the core theoretical derivations, and the experimental analyses were all carried out independently by the authors.

