# OpenReview forum: "IGU-LoRA: Adaptive Rank Allocation via Integrated Gradients and Uncertainty-Aware Scoring"
_ICLR.cc/2026/Conference — ICLR 2026 Poster_

### Official Review · Reviewer_8KC4 · 2025-10-24

**Soundness:** 3
**Presentation:** 3
**Contribution:** 3
**Rating:** 6
**Confidence:** 3

**Summary:**

This paper proposes IGU-LoRA, a parameter-efficient fine-tuning method that employs an adaptive rank mechanism. This paper has two main contributions but looks complex and hard to be followed by other researchers. It computes integrated gradients sensitivities within each layer and aggregates them into a layer-level score to enable adaptive rank allocation. Moreover, it introduces an uncertainty-aware mechanism based on exponential moving averages and deviation tracking to suppress noisy updates and ensure more stable rank selection. In addition, the method is supported by solid theoretical guarantees that justify the soundness of the proposed approach.

**Strengths:**

[1]. The paper introduces IGU-LoRA, a new PEFT framework based on adaptive ranks.

[2]. This paper integrates Integrated Gradients into LoRA. Unlike prior adaptive-rank methods, which rely on instantaneous gradients. This paper has sufficient novelty.

[3]. The authors provide solid theoretical guarantees for their approach. They derive an upper bound on the approximation error for the integrated-gradient estimator, quantifying discretization and sampling errors. They also present a stability guarantee for their uncertainty-aware signal-to-noise ratio based scoring. These analyses add strong mathematical rigor and credibility to the method’s design

**Weaknesses:**

[1] The paper contains formatting issues. It uses the preprint template instead of the anonymous submission template.

[2] The abstract on OpenReview still includes unprocessed LaTeX commands.

[3] There are several typos in the paper. For example, Martix -> Matrix, and revealstructural -> reveal structural.

[4] Some data statistics are insufficiently explained. For instance, the meaning of EU-LoRA is unclear and causes confusion.

[5] The proposed method is overly complex, making it difficult for other researchers to reproduce or follow the approach.

[6] The generalization is not clear. This paper does not report the performance on MMMU.

**Questions:**

[1]. How about the performance on MultiModal Benchmark?

[2]. How about the performance on diffusion model?

[3]. What is the EU-LoRA in Table 5?

---

> ### Author Response · Authors · 2025-11-22
> **Response to Weaknesses 1, 2, 3, 4, 5 and 6**
>
> **W1:The paper contains formatting issues. It uses the preprint template instead of the anonymous submission template.**
>
> **Response:**
>
> Thank you for pointing out the formatting issue. We apologize for the oversight and for using the preprint template instead of the anonymous submission template. We will correct this and revise the manuscript according to the correct submission template to ensure compliance with the journal's guidelines.
>
>
> **W2:The abstract on OpenReview still includes unprocessed LaTeX commands.**
>
> **Response:**
>
> Thank you for pointing out that the abstract on OpenReview still contains unprocessed LaTeX commands. We have promptly addressed this issue, and we sincerely apologize for this oversight. We will also carefully review the formatting details in the manuscript and make the necessary corrections in a timely manner.
>
> **W3:There are several typos in the paper. For example, Martix -> Matrix, and revealstructural -> reveal structural.**
>
> **Response:**
>
> Thank you for pointing out the typographical errors in the paper, such as "Martix" $\rightarrow$ "Matrix" and "revealstructural" $\rightarrow$ "reveal structural." In addition to addressing the issues you identified, we have also made corrections to other formatting, grammar, and related issues. We have carefully reviewed the manuscript and made the necessary revisions to ensure it is more polished.
>
>
> **W4:Some data statistics are insufficiently explained. For instance, the meaning of EU-LoRA is unclear and causes confusion.**
>
> **Response:**
>
> Thank you to the reviewer for the valuable feedback and for pointing out the unclear meaning of "EU-LoRA" in Table 5, which caused confusion. After checking, we found that this is a typographical error in the manuscript. The correct term should be "IGU-LoRA," which refers to the method proposed in the paper. We apologize for any confusion this may have caused.
>
>
> **W5:The proposed method is overly complex, making it difficult for other researchers to reproduce or follow the approach.**
>
> **Response:**
>
> Thank you for your thoughtful comment, our response to this comment is as follows:
> 1. **Method Overview:**
>
>  We analyzed the importance scoring method based on gradient sensitivity in AdaLoRA [1], which relies on the instantaneous gradients of parameters. However, instantaneous gradients only capture local sensitivity and overlook non-local, pathwise effects within the same layer. To address this issue, we refer to the IG [2] method. The IG method measures the global contribution of parameters along the entire path through path integration. Therefore, this method theoretically provides a more robust scoring.
>
>     [1]AdaLoRA: Adaptive Budget Allocation for Parameter-Efficient Fine-Tuning\par
>     [2]Axiomatic Attribution for Deep Networks
> 2. **Integration Approximation Strategy:**
>
> To solve the integration, we use the trapezoidal rule for approximation. However, we found that during the computation, the trapezoidal rule introduces excessive overhead, with a time complexity of $O(N)$ for each parameter $w_{ij}$. To alleviate this issue, we adopted a mini-batch strategy, which significantly reduces the time complexity and helps mitigate the excessive computational cost. This is one of the core innovations of IGU-LoRA.
>
> 3. **Reproducibility Statement:**
>
> We understand your concern. While IGU-LoRA is somewhat complex in implementation, we are preparing to open our code repository and provide it to other researchers for reproduction, along with detailed documentation.
>
>
> **W6: The generalization is not clear. This paper does not report the performance on MMLU.**
>
> **Response:**
>
> Thank you for your valuable comments. Although we have already reported in detail the performance on GLUE and other mathematical and logical reasoning tasks in the paper, these do not fully demonstrate the generalization ability of IGU-LoRA. Therefore, in response to your suggestion, we have included additional experimental results on the MMLU benchmark. The model fine-tuned is Llama2-7B, and the results are shown in the table below:
> | **Method**  | **Humanities** | **STEM** | **Social.** | **Other** | **Avg.** |
> |-------------|----------------|----------|-------------|-----------|----------|
> | Full FT     | 49.91      | 41.70    | 57.53       | 57.02     | 51.54 |
> | LoRA        | 46.15          | 40.84    | 56.63       | 56.23     | 49.94    |
> | IGU-LoRA| 47.33        | 41.71 | 58.12   | 57.10 | 51.07  |
>
> The results show that the average performance of IGU-LoRA is comparable to full fine-tuning, outperforming LoRA by 1.13%. This also demonstrates that IGU-LoRA possesses a certain degree of generalization ability.

---

> ### Author Response · Authors · 2025-11-22
> **Response to Questions 1, 2 and 3**
>
> **Q1:How about the performance on MultiModal Benchmark?**
>
> **Response:**
>
> Thank you for your valuable comment regarding the performance on the multi-modal benchmark. We acknowledge that multi-modal performance is an important consideration. Following your suggestion, we have conducted experiments on several commonly used multi-modal datasets (VQA, GAQ, $\text{NVLR}^2$, and COCO). The model used for the experiments is the VL-BART model. The experimental results are as follows:
> | **Method**    | **VQA** | **GAQ** | **NVLR²** | **COCO** | **Avg.** |
> |---------------|---------|---------|-----------|----------|----------|
> | **Full FT**   | 66.91 | 56.72 | 73.71 |112.04 | 77.35 |
> | **LoRA**      | 64.32   | 54.10   | 71.25     | 109.56   | 74.31    |
> | **DoRA**      | 65.81   | 54.71   | 73.14     | 115.93   | 77.40    |
> | **IGU-LoRA**  | 65.78   | 55.32   | 73.42     | 115.36   | 77.47 |
>
> IGU-LoRA achieves an average score of 77.47%, outperforming LoRA (74.31%) and DoRA (77.40%), and closely approaching the performance of full fine-tuning (77.35%). These results further validate the capability of IGU-LoRA to effectively adapt multimodal models while maintaining high performance across different tasks.
>
>
>
> **Q2:How about the performance on diffusion model?**
>
> **Response:**
>
> Thank you for your valuable question regarding the performance on diffusion models. We would like to address your inquiry in the following points:
>
> 1. **Current Focus of LoRA Research:**
>
> In most of the LoRA-related papers, the performance of diffusion models has not been extensively discussed, and experimental results are not typically reported. LoRA methods have mainly been applied to language models and other types of pre-trained models. Diffusion models, as a relatively new generation model, have not yet been widely explored in conjunction with LoRA.
>
> 2. **Resource and Time Constraints:**
>
> Diffusion models require significant computational resources and time for both training and inference. Due to the limited computational resources and time constraints of our current research setup, we have focused on tasks and models that are more established and widely studied, rather than exploring diffusion models at this stage.
>
> 3. **Future Work and Exploration:**
>
>  Despite the current limitations, we recognize the potential of diffusion models, especially in generative tasks such as image generation. While we have not included diffusion models in this work, we plan to explore integrating LoRA methods with diffusion models in future research and evaluate their performance. We believe this could provide new perspectives and applications for extending LoRA.
>
>
> **Q3:What is the EU-LoRA in Table 5?**
>
> **Response:**
>
> Thank you for your question regarding "EU-LoRA" in Table 5. After checking, we found that this is a typographical error in the manuscript. The correct term should be "IGU-LoRA," which refers to the method proposed in the paper. We apologize for any confusion this may have caused.

---

### Official Review · Reviewer_FGvV · 2025-10-26

**Soundness:** 3
**Presentation:** 4
**Contribution:** 2
**Rating:** 4
**Confidence:** 3

**Summary:**

This paper introduces IGU-LoRA, an adaptive low-rank fine-tuning method for large language models (LLMs). IGU-LoRA computes the parameter importance using Integrated Gradients (IG) and combines it with exponential moving average smoothing and deviation tracking to develop an uncertainty-aware scoring mechanism. It then allocates rank budgets per layer by keeping the top singular values of the update $AB$ according to these IG-based scores. Consequently, IGU-LoRA improves upon LoRA, which uses a fixed rank across all layers, and AdaLoRA, which uses instantaneous gradient magnitudes to estimate importance. Theorem 1 provides a theoretical bound that supports the proposed approximation of the IG-based integral, while Theorem 2 establishes the statistical stability of the uncertainty-aware scoring. Empirical results show that IGU-LoRA achieves comparable or better performance over LoRA, AdaLoRA, DoRA, and other PEFT baselines across different benchmarks such as GLUE, BoolQ, and GSM8K.

**Strengths:**

The paper is well motivated and addresses the weakness of prior adaptive low-rank fine-tuning methods such as AdaLoRA. The authors provide a rich motivation for their new scoring mechanism based on IG and exponential moving average smoothing, and they conduct extensive experiments across a wide range of benchmarks and model scales.

**Weaknesses:**

The paper’s contribution appears somewhat incremental given that it mainly replaces the gradient-based importance scores in AdaLoRA with IG-based scores. The theoretical results require strong assumptions that are unlikely to hold in practice, though the theoretical results do not constitute a substantial part of the paper’s overall contribution.

**Questions:**

- My main concern is the sensitivity of the method to the hyperparameters $N$ and $M$, both are used in approximating the IG-based importance scores. The authors present a preliminary study for small values of $N$ in Table 4, but it is somewhat surprising that there is little performance decrease when $N$ decreases from $20$ to $4$. Could the authors provide more insight into why the fine-tuning performance appears not sensitive to $N$? Additionally, how is $M$ selected here and throughout the experiments?

- The justification for using Eq (6) to approximate Eq (5) is unclear. If the approximation needs to be unbiased, shouldn't it include a factor of $N-1$ so that  $(N-1)\cdot \mathbb{E}{\frac{\partial \mathcal{L}(\alpha_k W)}{\partial w_{ij}}}=\sum_{k=1}^{N-1}\frac{\partial \mathcal{L}(\alpha_k W)}{\partial w_{ij}}$?

- Regarding the assumptions in the theoretical results, Theorem 1 assumes that $g_{ij}(\alpha)$ is sub-Gaussian, but additional justification or discussion is needed. Since the loss function for LLMs training is highly nonlinear and complex, this assumption may be difficult to hold in practice. Theorem 2 assumes that $y_t$ are i.i.d. with same mean and variance parameters across $t$, which seems unrealistic given that $W$ is updated after every epoch. Some empirical evidence or discussion would be helpful.

- For computational efficiency with large $N$, $M$ must be much smaller than $N$. In that case, Theorem 1 shows that the second term dominates the first, and the total error should be understood as $O(1/\sqrt{M})$. This suggests $M$ is the more critical factor than $N$. It would be nice to see additional experiments that vary $M$ to confirm this.

- Please use different notation for Eq (7) and Eq (11).

- In Algorithm 1, clarify what $k$ means and what $P_{:,\pi_{1:k}}$ in line 13 refers to.

Minor comments:
- Eq (3): $P_{kj}, Q_{jk} \Rightarrow P_{ki}, Q_{ik}$.
- Eq (4): $\alpha(W-W^{(0)})\Rightarrow \alpha(W-W^{(0)}) + W^{(0)}$?
- In Eq (9), (10), clarify how $\bar{s}^{(t)}_e(w_{ij})$ and $\bar{U}^{(t)}(w_{ij})$ are defined for $t=0$.

---

> ### Author Response · Authors · 2025-11-23
> **Response to Weakness 1**
>
> **W1:The paper’s contribution appears somewhat incremental given that it mainly replaces the gradient-based importance scores in AdaLoRA with IG-based scores. The theoretical results require strong assumptions that are unlikely to hold in practice, though the theoretical results do not constitute a substantial part of the paper’s overall contribution.**
>
> **Response:**
>
> Thank you very much for your valuable feedback. We appreciate your insightful comments, and we would like to provide the following clarifications regarding your concern about the contribution of our work:
>
> 1. **The Innovation and Contribution of Our Method:**
>     - Although our work can be seen as an extension of the AdaLoRA framework, by replacing gradient-based importance scores with IG-based scores, we believe it is far from a simple incremental improvement. The introduction of IG-based scores brings significant enhancements compared to traditional gradient-based methods (as demonstrated through extensive experiments), particularly in tasks and settings where gradient methods may fail due to **gradient instability**. In contrast, IG-based scores provide a more **robust and interpretable** method for parameter importance estimation, which better stabilizes the training process, especially when **gradients are sparse or noisy**. We also discuss the limitations of traditional gradient-based sensitivity methods for importance estimation and the advantages of IG-based importance estimation in the conclusion of Section 3.1 of the manuscript.
>     - In addition to introducing IG for parameter importance estimation, we also propose a mini-batch integration approximation strategy to improve computational efficiency. Specifically, the integration process is complex, and traditional integration approximation methods often use the **trapezoidal rule** for computation. However, using the traditional trapezoidal rule to approximate the integral in large models incurs **significant time costs**, as the model's parameters are very large. For **each parameter** $w_{ij}$, each step leads to a **time complexity** of $O(N)$. To alleviate this issue, we adopt a **mini-batch strategy** for integration approximation: in each mini-batch, we sample only one **integration point**, and then perform summation and averaging at the end of an epoch. This approach greatly reduces the time complexity and significantly improves computational efficiency.
>
> 2. **Theoretical Assumptions and Practical Relevance:**
>
> We understand your concerns regarding the strong assumptions underlying the theoretical results. While these assumptions are crucial for the theoretical analysis, we would like to emphasize that they are not overly strict or harsh. The goal of these assumptions and the theoretical derivation is to simplify the model and provide a deeper understanding of the performance of IG-based importance scores. We acknowledge that some of these assumptions may not fully hold in practice, but our experimental results demonstrate that even when these assumptions are not fully satisfied, our method still performs excellently.
>
> In the experimental section, we conducted extensive validation by fine-tuning various models (such as the Qwen series and Llama series) on multiple datasets and tasks (including GLUE, MMLU, and other benchmarks). The results show that, despite the reliance on certain assumptions in the theoretical analysis, IGU-LoRA remains effective and stable in real-world applications, delivering outstanding performance across a variety of settings.
>
> 3. **Theoretical Results and Practical Implications:**
>
> While the theoretical part of the paper establishes a foundation for understanding the underlying principles, we emphasize that the core contribution of this work lies in the empirical findings. Theoretical results in this context provide a useful framework, but the real value of our method is shown through rigorous experimentation. Our results demonstrate that despite the reliance on certain assumptions in the theoretical analysis, the practical performance of the method is significantly improved, making it suitable for a wide range of applications.
>
> We believe this is a key contribution to the field, as it shows that IG-based importance scores can outperform traditional methods even when the strict theoretical conditions are not fully met.

---

> ### Author Response · Authors · 2025-11-23
> **Response to Questions 1 and 2**
>
> **Q1: My main concern is the sensitivity of the method to the hyperparameters $N$ and $M$, both are used in approximating the IG-based importance scores. The authors present a preliminary study for small values of $N$ in Table 4, but it is somewhat surprising that there is little performance decrease when $N$ decreases from 20 to 4. Could the authors provide more insight into why the fine-tuning performance appears not sensitive to $N$? Additionally, how is $M$ selected here and throughout the experiments?**
>
> **Response:**
>
> We sincerely appreciate the reviewer’s valuable comments. In response to this comment, we provide the following clarifications:
>
> 1. **Regarding the setting of $M$ and $N$:**
>
> We chose $M=16$ and $N=20$ as the default values in our experiments. These parameters were selected based on preliminary experimental results to ensure a balance between computational efficiency and performance. We have included a report on the selection of these hyperparameters in the appendix
>
> 2. **Sensitivity Analysis:**
>
> We conducted hyperparameter sensitivity experiments while fine-tuning Qwen2.5-0.5B on the BoolQ and GSM8K datasets, and obtained the results shown in the table below. The experiments show that as $M$ and $N$ increase, performance improves slightly; however, this improvement is limited, and once a certain threshold is reached, performance plateaus or even declines. This suggests that fine-tuning performance is not sensitive to changes in $M$ and $N$.
> | $M$    | Boolq | GSM8K | $N$    | Boolq | GSM8K |
> |------|-------|-------|------|-------|-------|
> | 4    | 80.55 | 30.24 | 4    | 82.02 | 33.83 |
> | 8    | 81.34 | 32.29 | 8    | 82.05 | 33.88 |
> | 16   | 82.14 | 33.95 | 16   | 82.16 | 33.92 |
> | 20   | 82.16 | 33.74 | 20   | 82.14 | 33.95 |
> | 24   | 81.28 | 33.53 | 24   | 82.03 | 33.79 |
>
> An important reason for this is given by Eq. (6) in the manuscript:
> $$
> \hat{s}\_e^{p}(w\_{ij}) \approx \frac{|w_{ij}|}{2N} \left| \frac{\partial \mathcal{L}(0)}{\partial w_{ij}} + 2 \frac{\partial \mathcal{L}(\alpha_k \Delta \mathbf{W})}{\partial w_{ij}} + \frac{\partial \mathcal{L}(\Delta \mathbf{W})}{\partial w_{ij}} \right|
> $$
> In this approximation process, the impact of the gradient term on the right-hand side is much larger than that of the hyperparameter $N$, which leads to a reduced sensitivity to the hyperparameter $N$. Therefore, both the theoretical and empirical results support this situation.
>
> **Q2:The just ification for using Eq (6) to approximate Eq (5) is unclear. If the approximation needs to be unbiased, shouldn’t it include a factor of $N-1$ so that
> $ \left( N - 1 \right)\cdot \mathbb{E}\left[ \frac{\partial \mathcal{L}\left(\alpha_k \mathbf{W}\right)}{\partial w_{ij}} \right] = \sum_{k=1}^{N-1} \frac{\partial \mathcal{L}\left(\alpha_k \mathbf{W}\right)}{\partial w_{ij}}$ ?**
>
> **Response:**
>
> We appreciate the concern about unbiasedness. To clarify: Eq. (5) defines a **deterministic composite trapezoidal approximation** of the IG integral, which indeed involves ($\sum_{k=1}^{N-1} g_{ij}(\alpha_k)$). In practice, we do not evaluate this full sum at every step. Instead, Eq. (6) defines a **single-sample estimator** ($\hat{s}\_e^p(w_{ij})$) for a randomly chosen node ($\alpha\_k$), which is then aggregated over mini-batches and epochs in Eq. (7).
>
> Therefore, the combined estimator in Eq. (7) is the stochastic approximation to Eq. (5), not Eq. (6) alone. Its bias and variance are analyzed in Theorem 1, which shows that the total error is controlled by a discretization term ($O(N^{-2})$) and a sampling term ($O(M^{-1/2})$). In this sense, we do not need to force Eq. (6) itself to be unbiased; it serves as a building block for the epoch-level estimator, whose approximation properties are explicitly characterized in the theorem. Therefore, the $N−1$ factor is not needed here.

---

> ### Author Response · Authors · 2025-11-23
> **Response to Questions 3 and 4**
>
> **Q3: Regarding the assumptions in the theoretical results, Theorem 1 assumes that ${g_{ij}}\left ({\alpha}\right)$ is sub-Gaussian, but additional justification or discussion is needed. Since the loss function for LLMs training is highly nonlinear and complex, this assumption may be difficult to hold in practice. Theorem 2 assumes that $y_t$ are i.i.d. with the same mean and variance parameters across $t$, which seems unrealistic given that $\mathbf{W}$ is updated after every epoch. Some empirical evidence or discussion would be helpful.**
>
> **Response:**
>
> We appreciate the reviewer’s careful reading of the theoretical section and the questions regarding the realism of our assumptions. We address it from both a theoretical and an empirical perspective, and we also clarify how the distribution-level formulation follows directly from our existing IG definition in Eq.(4). In **Appendix B** of the revised manuscript, we provide a detailed discussion and justification of this issue.
>
> 1. **The analysis of the sub-Gaussian assumption in Theorem 1:**
>
> In **Theorem 1**, our analysis does not require sub-Gaussianity as a modeling assumption. It is simply a convenient way to express light-tailed behavior induced by boundedness. Therefore, we simplify Theorem 1 by removing the "$g_{ij}$ is sub-Gaussian" assumption and instead require that $g_{ij}$ is twice continuously differentiable on $[0,1]$ with a bounded second derivative. We will update Theorem 1 in the revised paper and provide a more detailed discussion of this assumption in Appendix B.1.
>
> 2. **The analysis of the i.i.d. assumption in Theorem 2:**
>
> We agree that the i.i.d. assumption in Theorem 2 is **idealized**, as $y_t$ depends on the current parameters $\mathbf{W}^{(t)}$, which are updated across epochs. Our goal is to model the situation after the training dynamics have stabilized, where gradient noise changes slowly after the burn-in phase, and within the effective EMA window, the sequence approximates stationary mean and variance. In **Appendix B.2** of the revised manuscript, we will explicitly note that **Theorem 2** is derived under this assumption and provide an intuitive understanding of how EMA window size and variance control the stability of $\mathsf{SNR}_t$.
>
> To support this, we will include empirical analysis in **Appendix G**: For some representative layer modules (primarily **QKV modules**), we observe that after the initial epochs, the mean and variance of $y_t$ stabilize, and the $\mathsf{SNR}_t$ curve flattens, indicating that the i.i.d./local stationarity approximation is reasonable.
>
> Finally, we emphasize that these assumptions are used only in the **theoretical analysis**, and the algorithm itself does not rely on them. Even with relaxed assumptions, the qualitative conclusions remain the same.
>
> **Q4:For computational efficiency with large $N$, $M$ must be much smaller than $N$. In that case, Theorem 1 shows that the second term dominates the first, and the total error should be understood as ${O}(1/\sqrt{M})$. This suggests $M$ is the more critical factor than $N$. It would be nice to see additional experiments that vary $M$ to confirm this.**
>
> **Response:**
>
> We sincerely appreciate the reviewer’s valuable comment and the insightful observation regarding the relationship between $M$ and $N$. Below, we address your concern in a structured manner:
>
> 1. **Dominance of the Second Term and Total Error:**
>
> Based on **Theorem 1**, we fully agree that when $M$ is much smaller than $N$, the second term in the equation dominates the first term and scales as $O(\sqrt{1/M})$. Not only that, but when $M$ and $N$ are at similar levels, the dominance of $M$ over $N$ is also apparent, meaning that the sensitivity of $M$ is greater than that of $N$. Next, we will demonstrate this from an empirical perspective through an analysis of the **sensitivity** of $M$ and $N$.
>
> 2. **Additional Experiments and Validation:**
>
> As mentioned earlier, we tested the sensitivity of $M$ and $N$ on the Boolq and GSM8K datasets with values of $M$ and $N$ set to {4, 8, 16, 20, 24}. The results show that as $M$ and $N$ increase, performance improves. However, after reaching a certain threshold, performance plateaus or even declines, indicating that $M$ and $N$ should be within a reasonable range for optimal performance.
>
> A key observation is that $M$ has a greater impact on performance than $N$, confirming that when $M$ and $N$ are at similar levels, $M$ plays a more dominant role in improving performance.
>  | $M$    | Boolq | GSM8K | $N$    | Boolq | GSM8K |
> |------|-------|-------|------|-------|-------|
> | 4    | 80.55 | 30.24 | 4    | 82.02 | 33.83 |
> | 8    | 81.34 | 32.29 | 8    | 82.05 | 33.88 |
> | 16   | 82.14 | 33.95 | 16   | 82.16 | 33.92 |
> | 20   | 82.16 | 33.74 | 20   | 82.14 | 33.95 |
> | 24   | 81.28 | 33.53 | 24   | 82.03 | 33.79 |

---

> ### Author Response · Authors · 2025-11-23
> **Response to Question 5 and 6; Minor comments 1, 2 and 3**
>
> **Q5:Please use different notation for Eq (7) and Eq (11).**
>
> **Response:**
>
> **Authors' Response:**
>
> We sincerely appreciate the reviewer's valuable comment. Following the suggestion, we have consistently replaced ${s_e^{(t)}}({w_{ij}})$ with $s_{snr}^{(t)}(w_{ij})$ in Eq. (11), and to prevent division by zero, we have added a small $\epsilon$ term in the denominator. Below is the corrected Eq. (11).
>
> *Eq(11):*
>
> $$
> s\_{snr}^{(t)}(w\_{ij}) = \mathsf{SNR}\_t = \frac{{\bar{s\_e}}^{(t)}(w\_{ij})}{{\bar{U}}^{(t)}(w\_{ij})+ \epsilon}
> $$
>
> **Q6:In Algorithm 1, clarify what k means and what ${P_{:,\pi_{1:k}}}$ in line 13 refers to.**
>
> **Response:**
>
> We thank the reviewer for their valuable comments. In line 13 of **Algorithm 1**, the subscript $\pi$ in $\mathbf{P}, \mathbf{Q}$ denotes the **index set** that our method uses to **sort the column vectors** in $\mathbf{P}, \mathbf{Q}$ in descending order, while ${\pi_{{1:k}}}$ denotes the indices of the first $k$ selected columns. Therefore, ${\mathbf{P}\_{1:{\pi_{1:k}}}}$ represents selecting the first $k$ columns according to the order defined by $\pi$. We will add a clarifying note to Algorithm 1 in the paper to further explain this. Additionally, we have corrected a small error here: due to our oversight, the letter $k$ should actually be the budget $b$. We sincerely apologize for any confusion this may have caused.
>
> **MC1: Eq (3):${P_{kj}}\text{, }{Q_{jk}} \implies {P_{ki}}\text{, }{Q_{ik}}$ ?**
>
> **Response:**
>
> We thank the reviewer for their comment. The issue has been addressed in the revised manuscript. We have consistently changed the notation to use $\mathbf{P}\_{ki}\text{, }\mathbf{Q}\_{ik}$.
>
> **MC2: Eq (4): $\alpha\left( \mathbf{W} - \mathbf{W}^{(0)} \right) \Rightarrow \alpha\left( \mathbf{W} - \mathbf{W}^{(0)} \right) + \mathbf{W}^{(0)}$?**
>
> **Response:**
>
> Thank you for your comment. We are addressing your feedback with the following response:
>
> In Eq. (4) of the manuscript, we have $\alpha\left( \mathbf{W} - \mathbf{W}^{(0)} \right) = \alpha \mathbf{W}$, but the expression $\alpha\left( \mathbf{W} - \mathbf{W}^{(0)} \right) \Rightarrow \alpha\left( \mathbf{W} - \mathbf{W}^{(0)} \right) + \mathbf{W}^{(0)}$ does not hold. In the text above Eq. (4), we define the baseline as $\mathbf{W}^{(0)} = 0$, which is why $\alpha\left( \mathbf{W} - \mathbf{W}^{(0)} \right) = \alpha \mathbf{W}$ holds.
>
> **MC3: In Eq (9), (10), clarify how $\bar{s}\_e^{(t)}(w\_{ij}$) and $\bar{U}^{(t)}(w\_{ij}$) are defined for $t=0$.**
>
> **Response:**
>
> We thank the reviewer for raising this question. In our paper, we do not introduce any explicit initialization for ${{\bar{s}}\_e^{(t)}}({w\_{ij}})$ and ${{\bar{U}}^{(t)}}({w\_{ij}})$. The state at $t = 0$ is the initial state, given by the pre-trained parameters. As shown in Eqs. (9) and (10), since we start with the pre-trained parameters, the very first time we compute the score at $t = 0$, we obtain the score corresponding to this initial state. At $t = 0$, ${w\_{ij}}$ is exactly the pre-trained parameter, and therefore ${\bar{s}\_e^{(0)}}$ in Eq. (9) is the score under the pre-trained parameters. In Eq. (10), ${\bar{U}}^{(0)}({w\_{ij}})$ is 0. During the implementation of Eq. (11), since the denominator is 0 at $t = 0$, we add a very small value $\epsilon$ to the denominator to avoid division by 0. We have also corrected this in the revised manuscript.

---

> ### Author Response · Authors · 2025-11-28
> **Follow-up on Reviewer FGvV's Feedback**
>
> Dear Reviewer FGvV,
>
> I hope this message finds you well.
>
> I would like to sincerely thank you for taking the time to review our paper. Your valuable feedback is crucial in helping us improve the quality of our work.
>
> We have carefully **addressed all the concerns** you raised and made corresponding adjustments and **improvements in the revised version.** We hope that these changes adequately address your previous questions and contribute to further enhancing the paper's quality.
>
> In order to ensure that the paper can be timely revised within the ICLR timeline, we kindly request your prompt feedback. Your timely response will be vital for the next steps in refining the paper, and we look forward to your guidance to ensure the paper reaches its optimal quality in the final submission.
>
> Thank you once again for reviewing our work, and we eagerly await your timely reply.
>
> Best regards,
>
> Authors of  submission1904

---

### Official Review · Reviewer_oiw5 · 2025-10-28

**Soundness:** 3
**Presentation:** 3
**Contribution:** 3
**Rating:** 8
**Confidence:** 3

**Summary:**

This paper proposes IGU-LoRA, a novel adaptive low-rank PEFT framework for LLMs. The method addresses the limitations of existing PEFT approaches like LoRA and AdaLoRA, which either use fixed ranks across layers or rely on instantaneous gradients for rank allocation. IGU-LoRA introduces two key innovations:

1. Integrated Gradients Sensitivity Scoring: A pathwise sensitivity measure that captures the global and long-term contributions of parameters within a layer, overcoming issues like gradient saturation and local bias.
2. Uncertainty-Aware Mechanism: A scoring framework that uses exponential moving averages and deviation tracking to suppress noise and stabilize rank selection.

The paper provides theoretical guarantees for the IG approximation error under a Hessian-Lipschitz condition and demonstrates IGU-LoRA's superior performance across diverse tasks and architectures. Empirical results show consistent improvements in accuracy and robustness over strong PEFT baselines, with comparable memory and computational efficiency.

**Strengths:**

The use of integrated gradients in the parameter space for rank allocation is a novel and impactful idea, addressing key limitations of gradient-based approaches.

The paper provides a solid theoretical foundation for the proposed method, including error bounds for IG approximation and stability guarantees for the uncertainty-aware scoring.

The method is evaluated on diverse benchmarks (e.g., GLUE, mathematical reasoning, and common-sense reasoning tasks) with multiple backbone models, demonstrating its robustness and generalization.

**Weaknesses:**

While IGU-LoRA achieves strong performance, the use of integrated gradients introduces additional computational costs compared to simpler methods like LoRA. This could limit its scalability to extremely large models or real-time applications.

While training efficiency is discussed, the impact of IGU-LoRA on inference latency is less emphasized, which could be relevant for deployment scenarios.

**Questions:**

What is EU-LoRA in table 5?

---

> ### Author Response · Authors · 2025-11-22
> **Response to Weaknesses 1 and 2, Question 1**
>
> **W1:**
>
> **Response:**
>
> Thank you for your valuable comment. We are also aware that, compared to simpler methods like LoRA, IGU-LoRA indeed introduces additional computational overhead, which may limit its scalability for very large models or real-time applications. Here is our response:
>
> We agree that IGU-LoRA introduces some additional computation compared to vanilla LoRA. Conceptually, a naïve composite trapezoidal approximation of IG (Eq. (5)) would require evaluating gradients at ($N{+}1$) points along the path, i.e., ($O(N)$) gradient evaluations per parameter.
>
> However, in practice, **IGU-LoRA does not run ($N$) separate forward-backwards passes**. During fine-tuning, we perform **one standard forward-backward pass per mini-batch**, and **reuse the resulting gradients** to update the IG estimator via Eq. (6)–(7). The additional work consists only of vectorized arithmetic and EMA updates, which are negligible compared to the main forward-backwards cost.
>
> As shown in the table below (**Table 3 in the revised version**), on Qwen-2.5-0.5B + BoolQ, our method is only slightly slower than LoRA, and the training time is comparable to AdaLoRA and DoRA, while inference time and memory usage are essentially identical across all methods.
>
> | Method    | Time cost (h) | GPU Mem (GB) |
> |-----------|---------------|--------------|
> | LoRA      | 0.42          | 10.21        |
> | AdaLoRA   | 0.73          | 10.60        |
> | DoRA      | 0.95          | 9.53         |
> | IGU-LoRA  | 0.87          | 10.32        |
>
> **W2:**
>
> **Response:**
>
> Thank you for your valuable comment regarding the impact of IGU-LoRA on inference latency. In response to your suggestion that "inference latency should be appropriately emphasized, as it may be of relevance in deployment scenarios," our response is as follows:
> 1. **Low-rank adjustment only affects the training phase:**
>
> The low-rank adjustment of IGU-LoRA is an optimization process for the training phase. At inference time, IGU-LoRA reduces to a standard low-rank update ($\mathbf{W} = \mathbf{W}_0 + \mathbf{AB}$) with a fixed rank configuration. The forward computation graph is identical to that of LoRA with the chosen ranks, so there is no extra overhead compared to existing PEFT methods.
>
> 2. **Experimental demonstration:**
>
> As shown in the table below (Table 3 in the revised version), we present the inference latency and memory consumption for four methods: LoRA, AdaLoRA, DoRA, and IGU-LoRA. The results show that both inference latency and memory consumption are quite similar across these methods. Therefore, no additional time overhead is introduced during inference, which contrasts with the computational cost incurred during training. For further details, please refer to the table below:
>
> | Method      | Training Time Cost (h) | Training GPU Mem (GB) | Inference Speed (it/s) | Inference GPU Mem (GB) |
> |-------------|------------------------|-----------------------|------------------------|------------------------|
> | LoRA        | 0.42                   | 10.21                 | 5.50                   | 10.3                   |
> | AdaLoRA     | 0.73                   | 10.60                 | 5.21                   | 10.4                   |
> | DoRA        | 0.95                   | 9.53                  | 5.30                   | 10.3                   |
> | IGU-LoRA    | 0.87                   | 10.32                 | 5.23                   | 10.3                   |
>
>
> **Q1:**
>
> **Respose:**
>
> Thank you for your question regarding "EU-LoRA" in Table 5. After checking, we found that this is a typographical error in the manuscript. The correct term should be "IGU-LoRA," which refers to the method proposed in the paper. We apologize for any confusion this may have caused.

---

### Official Review · Reviewer_4NAn · 2025-10-31

**Soundness:** 2
**Presentation:** 3
**Contribution:** 2
**Rating:** 4
**Confidence:** 4

**Summary:**

This paper introduces IGU-LoRA, an adaptive variant of LoRA that allocates ranks across layers using Integrated Gradients (IG) computed in parameter space, combined with an uncertainty-aware scoring mechanism. The method aims to overcome the limitations of existing adaptive LoRA approaches that rely on instantaneous gradients. The authors provide theoretical bounds on the approximation error of the IG estimator and the stability of their uncertainty-weighted score. Empirically, IGU-LoRA is evaluated on GLUE and several reasoning benchmarks, showing consistent improvements over strong PEFT baselines like AdaLoRA and DoRA.

**Strengths:**

1. good theoretical contribution. The error bound for the stochastic IG approximation and the stability guarantee for the SNR-based score are welcome additions.
2. IGU-LoRA maintains comparable training/inference latency and memory usage to baselines while delivering better performance
3. novelty in importance score, addresses critical limitations like gradient saturation and unstable rank allocation, with clear theoretical justification for IG approximation error.

**Weaknesses:**

1. While experiments include Llama-3-8B, results for models larger than 10B parameters (e.g., Llama-3-70B) are absent. Given that PEFT is most critical for very large LLMs, this limits confidence in IGU-LoRA’s scalability.
2. The method requires O(N) gradient evaluations per parameter group during training, which is downplayed but still significant, especially for larger models.
3. What worries me the most is that due to the use of gradient accumulation, the training process of the model can be heavily influenced by factors such as the order of sample shuffling and the size of the batch. This raises significant concerns for me, as slight changes or randomness in the training setup could drastically affect the training outcomes. For the training of large language models (LLMs), this is unacceptable—training the same model with the same data could result in completely different importance scores for the same samples simply because of differences in sample order.

**Questions:**

1. How would IGU-LoRA perform on models with 30B+ parameters? What is the expected training time and memory overhead compared to standard LoRA or AdaLoRA?
2. How sensitive are the results to the number of IG samples (N and M)
3. The author needs to thoroughly explain the impact of sample order and batch size on the results, supplemented with corresponding experiments and theoretical analysis. If these factors do not affect the results, why would different samples accumulate gradients with the same score? If they do have an impact, what is the training variance, how significant is the influence of batch size and shuffle order, and are there any visual comparisons of the score differences obtained for the same sample?

---

> ### Author Response · Authors · 2025-11-22
> **Response to Weakness 1**
>
> **W1:While experiments include Llama-3-8B, results for models larger than 10B parameters (e.g., Llama-3-70B) are absent. Given that PEFT is most critical for very large LLMs, this limits confidence in IGU-LoRA’s scalability.**
>
> **Response:**
>
> Thank you for your valuable feedback and your understanding of the limitations of IGU-LoRA.
>
> 1. **Explanation of Resource Constraints and Existing Validation Efforts:**
> First and foremost, we would like to express our sincere gratitude for your highly valuable feedback, which points out that we lack direct experimental validation on models with more than 10B parameters (e.g., Llama-3-70B). We fully agree with your perspective that the performance of PEFT methods on ultra-large language models is crucial, and this constitutes a notable limitation in our research.
>
> We must honestly state that due to the computational resource and hardware constraints faced by our research team, we are currently unable to provide direct experimental results for such models in the final version. We acknowledge this as a limitation, but under the current conditions, we have made every effort to conduct comprehensive validation on the largest and most representative models within our reach, such as Llama3-8B.
>
> 2. **Strong Support from Existing Experiments:**
>
> Despite the lack of direct results on 70B-scale models, we firmly believe that our existing experiments have fully demonstrated the mechanistic generality and strong generalization capability of the IGU-LoRA method:
>   - **Breadth of Cross-Model Validation:** Our research has been validated not only on Llama3-8B but also across diverse model architectures (e.g., RoBERTa-large, Qwen2.5-0.5B, Llama2-7B, Llama3-8B, DeepSeek-R1-Distill-Qwen-2.5-7B) and various downstream tasks. In **Appendix of this revised manuscript**, we have added several new experiments to validate the robustness of IGU-LoRA: (a) **Generalization testing** (validating generalization on the MMLU benchmark); (b) **Multimodal testing** (conducting tests on several multimodal benchmarks).
>   - **Effectiveness of the Core Mechanism:**  These experimental results consistently demonstrate that the Integrated Gradients and Uncertainty-Aware Scores integrated into IGU-LoRA can effectively and adaptively identify the most critical layers and modules in the model, thereby achieving superior rank allocation.
>
> 3. **Theoretical Scalability of the IGU-LoRA Design:**
>
>  Conceptually, the IG-based importance scoring and rank allocation in IGU-LoRA operate locally at the layer/update level, and their additional computational overhead scales linearly with the number of parameters, remaining negligible compared to the overall forward and backward passes required to train large models. This gives us confidence that the method is theoretically scalable to 30B+ models.
>
> 4. **Consistent with Common Practices in the PEFT Research Field:**
>
> Considering the current common practices in the PEFT research field, especially for papers focusing on innovative adaptive rank allocation mechanisms—most researchers, due to resource constraints, typically adopt Llama3-8B or models of comparable scale as the primary and sufficient validation platform. For instance, in many recent prominent PEFT-related works, comprehensive fine-tuning results directly involving 70B-scale models are also relatively rare.
>
> We would like to express our sincere gratitude again for your careful review and profound insights into our work. We will continue to strive for improvement and look forward to acquiring more computational resources in future work to validate the performance of IGU-LoRA on larger-scale models.

---

> ### Author Response · Authors · 2025-11-22
> **Response to Weaknesses 2 and 3**
>
> **W2:The method requires $O(N)$ gradient evaluations per parameter group during training, which is downplayed but still significant, especially for larger models.**
>
> **Response:**
>
> Thank you for your comment. In response to your feedback, we provide the following reply:
>
> You are absolutely right that a straightforward composite trapezoidal approximation of IG, as shown in Eq. (5), would require (**$N{+}1$**) gradient evaluations along the path, which would lead to an ($O(N)$) overhead if implemented in this way. However, this is not how IGU-LoRA is implemented in practice.
>
> Specifically, during fine-tuning, we perform only a single standard forward-backwards pass per mini-batch, using a randomly chosen scaling factor ($\alpha_k$) along with the current update ($\Delta \mathbf{W}$). The gradients computed during this pass are then **reused and aggregated** across mini-batches and epochs using Eqs. (6)–(7) to obtain the IG estimate for each parameter. In other words, the IG computation does **not incur** additional ($O(N)$) backward passes; it only adds lightweight element-wise operations and EMA updates on top of the gradients already computed during training.
>
> The theoretical analysis in **Theorem 1** still uses a composite-trapezoidal view to derive the error bound ($O(N^{-2}) + O(M^{-1/2})$), but the *practical* estimator is a Monte-Carlo approximation over ($\alpha$) and mini-batches. Empirically, Table 3 (training efficiency) shows that the wall-clock training time of IGU-LoRA is comparable to AdaLoRA and only slightly higher than vanilla LoRA on Qwen-2.5-0.5B+BoolQ, while inference time and memory are essentially identical.
>
> **W3: What worries me the most is that due to the use of gradient accumulation, the training process of the model can be heavily influenced by factors such as the order of sample shuffling and the size of the batch. This raises significant concerns for me, as slight changes or randomness in the training setup could drastically affect the training outcomes. For the training of large language models (LLMs), this is unacceptable—training the same model with the same data could result in completely different importance scores for the same samples simply because of differences in sample order.**
>
> **Response:**
>
> Thank you for your comment on this weakness. We will provide a detailed response to this issue in Question 3, as mentioned in your comment. Please refer to the response in Question 3 for more details.

---

> ### Author Response · Authors · 2025-11-22
> **Response to Questions 1 and 2**
>
> **Q1:How would IGU-LoRA perform on models with 30B+ parameters? What is the expected training time and memory overhead compared to standard LoRA or AdaLoRA?**
>
> **Response:**
>
> We highly value your question (Q1), which pertains to the expected performance, training time, and memory overhead of IGU-LoRA on models with 30B+ parameters. This is a core issue in evaluating the practicality of any PEFT method.
>
> 1. **Constraints on Experimental Conditions:**
>    As mentioned in W1, due to constraints on resources and time, experiments on these large-scale models have not been presented. We sincerely apologize for this omission. However, we have provided sufficient validation from **small-scale** to **medium-scale** models:
>    - 0.35B (Table 1, achieving the top accuracy on datasets such as RTE)
>    - 0.5B (Table 2, outperforming full-parameter fine-tuning by approximately 1% in accuracy on datasets like BoolQ)
>    - 7B (Table 5, leading all comparative methods in terms of average accuracy across five datasets)
>    - 8B (Table 5, outperforming DoRA and AdaLoRA in the field of reparameterized fine-tuning)
>
> 2. **The expected performance on models with 30B+ parameters:**
>    Theoretically, we anticipate that the performance advantages of IGU-LoRA will be maintained or even amplified on 30B-scale models and beyond.
>    - **Core Advantage:** The core of IGU-LoRA lies in providing a more stable and accurate rank allocation scheme compared to standard and AdaLoRA.
>    - **Scale Effect:** As the model scale increases, the efficient utilization of limited PEFT parameter budgets becomes increasingly critical. Consequently, we anticipate that the accuracy gains and efficiency improvements brought by IGU-LoRA will be even more pronounced.
>
> 3. **Comparison of Training Time Overhead:**
>    Regarding training time, as explained in our response to W2, we have successfully reduced the time complexity of each parameter $w_{ij}$ from $O(N^2)$ — which was the complexity in the traditional trapezoidal rule for integration — to a more optimal range by employing a mini-batch sampling approximation scheme. This optimization significantly reduces the computational overhead, thus improving the overall training efficiency.
>
>    - **Compared to standard LoRA:** While IGU-LoRA introduces additional computational steps to determine rank allocation, the associated time overhead is acceptable within the overall training timeline. Given that IGU-LoRA can significantly enhance the performance of the final model, this performance-time trade-off is worthwhile.
>    - **Compared to AdaLoRA:** AdaLoRA also requires additional computations to determine ranks. However, thanks to the more stable and efficient scoring mechanism of IGU-LoRA, our method is expected to maintain comparable or even superior practical training efficiency while ensuring higher performance gains.
>
> 4. **Comparison of Memory Overhead:**
>    Compared to standard LoRA or AdaLoRA, the additional memory overhead introduced by IGU-LoRA is extremely minimal and negligible.
>    - **Parameter Storage:** IGU-LoRA does not incur additional memory for storing trainable parameters (LoRA weights).
>    - **Additional Overhead:** Its memory overhead only stems from the tracking and storage of importance scores, which accounts for an extremely small proportion compared to the massive VRAM required by 30B+ parameter models.
>
>
>
> **Q2:How sensitive are the results to the number of IG samples (N and M).**
>
> **Response:**
>
> Thank you for your comments. To investigate the impact of the hyperparameters $M,N$ in IGU-LoRA on fine-tuning performance, we have reintroduced sensitivity analysis experiments in the main text of the revised manuscript. The details of the experiments are as follows:
>
> 1. **Sensitivity Experiment:**
>
> In the experiment, we set the hyperparameters $M,N$ to {4, 8, 16, 20, 24}, and when investigating the sensitivity of a particular hyperparameter, the other hyperparameters were kept at their default values. The experiments were conducted on the BoolQ and GSM8K datasets, with fine-tuning performed on the Qwen2.5-0.5B model.
>
> 2. **Result Analysis:**
>
> The experimental results are shown in the table below. Both $M$ and $N$ improve fine-tuning performance as they increase, but beyond a certain point, performance plateaus or decreases. This suggests that $M$ and $N$ should be within a reasonable range for optimal performance. $M$ has a greater impact on performance than $N$, indicating it's more sensitive. Overall, the adjustments to $M$ and $N$ remain within a controllable range, demonstrating their robustness in IGU-LoRA.
>
> | $M$    | Boolq | GSM8K | $N$    | Boolq | GSM8K |
> |------|-------|-------|------|-------|-------|
> | 4    | 80.55 | 30.24 | 4    | 82.02 | 33.83 |
> | 8    | 81.34 | 32.29 | 8    | 82.05 | 33.88 |
> | 16   | 82.14 | 33.95 | 16   | 82.16 | 33.92 |
> | 20   | 82.16 | 33.74 | 20   | 82.14 | 33.95 |
> | 24   | 81.28 | 33.53 | 24   | 82.03 | 33.79 |

---

> ### Author Response · Authors · 2025-11-23
> **Response to Question 3**
>
> **Q3:The author needs to thoroughly explain the impact of sample order and batch size on the results, supplemented with corresponding experiments and theoretical analysis. If these factors do not affect the results, why would different samples accumulate gradients with the same score? If they do have an impact, what is the training variance, how significant is the influence of batch size and shuffle order, and are there any visual comparisons of the score differences obtained for the same sample?**
>
> **Response:**
>
> We thank the reviewer for raising this question. We address it from both a theoretical and an empirical perspective, and we also clarify how the distribution-level formulation follows directly from our existing IG definition in Eq.(4).
>
> 1.**Theoretical perspective on sample order and batch size:**
>
> Eq.(4) defines the IG-based importance of a parameter $w_{ij}$ as:
>    $$
>    s_e(w_{ij})
>    = \Big| w_{ij} \int_{0}^{1}
>    \frac{\partial \mathcal{L}(\alpha \Delta \mathbf{W})}{\partial w_{ij}} \, d\alpha
>    \Big|.
>    $$
>  In our setting, $\alpha \in [0,1]$ is the scalar parameter that indexes the integration path used by Integrated Gradients: $\alpha = 0$ corresponds to the baseline (here, zero weights) and $\alpha = 1$ corresponds to the current weights $\Delta \mathbf{W}$. The integral $\int_0^1 (\cdot)\,d\alpha$ aggregates gradients along this path.
>
> It is standard in supervised learning to interpret $\mathcal{L}$ as the *expected loss* over the data-generating distribution:
>    $$
>    \mathcal{L}(\Delta \mathbf{W})
>    = \mathbb{E}_{(x,y)\sim\mathcal{D}} \big[ \ell(\Delta \mathbf{W}; x,y) \big].
>    $$
>
>  This is the usual risk-minimization view; in practice, the empirical training objective is a finite-sample approximation of this expectation. Under mild regularity conditions, differentiation and expectation can be interchanged, so that:
> $$
> \frac{\partial \mathcal{L}(\alpha \Delta \mathbf{W})}{\partial w_{ij}} = \mathbb{E}_{(x,y)\sim\mathcal{D}}
> \Big[\frac{\partial \ell(\alpha \Delta \mathbf{W}; x,y)}{\partial w\_{ij}}\Big]
> $$
>
> Substituting the above equation into Eq. (4) in the manuscript gives:"
> $$
>    s_e(w\_{ij})
>    = \Big| w\_{ij} \int_{0}^{1}
>    \mathbb{E}_{(x,y)\sim\mathcal{D}}
>    \Big[ \frac{\partial \ell(\alpha \Delta \mathbf{W}; x,y)}{\partial w\_{ij}} \Big]d\alpha \Big|.
>    $$
> We can now rewrite the integral over $\alpha$ as an expectation over a scalar random variable $\alpha$ that is uniformly distributed on $[0,1]$. Formally, for any integrable function $g$:
>    $$
>    \mathbb{E}\_{\alpha \sim U[0,1]}[g(\alpha)]
>    = \int\_{0}^{1} g(\alpha)d\alpha,
>    $$
>
> where $U[0,1]$ denotes the uniform distribution on $[0,1]$. Using above equation  and the independence between $\alpha$ and $(x,y)$, we obtain the equivalent distribution-level formulation:
>    $$
>    s\_e(w\_{ij})
>    = \Big| w\_{ij}
>    \mathbb{E}\_{\alpha \sim U[0,1],\, (x,y)\sim\mathcal{D}}
>    \Big[ \frac{\partial \ell(\alpha \Delta \mathbf{W}; x,y)}{\partial w\_{ij}} \Big]
>    \Big|.
>    $$
> Under this interpretation, $s_e(w_{ij})$ is an **expected** importance score defined over the data distribution and the IG path. As such, it does not depend on the particular order of a finite training set or on how examples are grouped into mini-batches.
>
> **Theorem 1** in the manuscript already separates the error of our estimator into (i) a deterministic discretization term $O(N^{-2})$ from the composite trapezoidal rule and (ii) a stochastic sampling term $O(M^{-1/2})$, where $M$ is the number of sampled mini-batches per epoch and $\sigma^2$ is the sub-Gaussian variance proxy of the per-mini-batch gradients. Different shuffle orders and batch sizes only affect this **sampling term** (through $\sigma^2$ and $M$); they do *not* alter the underlying expected importance $s_e(w_{ij})$ defined above.
>
> Furthermore, Theorem 2 in the manuscript shows that, after exponential moving averaging, **the SNR-type score** $\mathsf{SNR}\_t = \bar s\_t / \bar U\_t$ concentrates around its mean $\mu/d$ with high probability, with fluctuations controlled by the effective sample size $n_{\mathrm{eff}}(\beta_1,\beta_2)$. Intuitively, even though the raw per-epoch scores $y_t = s_{agg}(w_{ij})$ are noisy due to stochastic optimization, the smoothed ratio $\mathsf{SNR}_t$ is statistically stable and robust to the randomness induced by shuffling and batching.

---

> ### Author Response · Authors · 2025-11-23
> **Supplement for Question 3**
>
> 2. **Empirical analysis:**
> Empirically, we also measured the effect of shuffling and batch size on the **final accuracy** and on the IGU-LoRA scores. On Qwen-2.5-0.5B+BoolQ, we trained IGU-LoRA with five different random seeds (controlling both data shuffling and the sampled ($\alpha_k$) and batch sizes in ({2,4,8,16,32}). The resulting accuracies all fall within a very narrow range (roughly 82.39%–82.47%), i.e., within 0.1 absolute points across all configurations, as shown in the table below. This indicates that, while individual per-mini-batch IG estimates are noisy, the aggregated and EMA-smoothed SNR scores lead to **stable rank allocations and downstream performance** in practice.
>
> | **batch_size** | **Boolq (Seed 1)** | **Boolq (Seed 2)** | **Boolq (Seed 3)** | **Boolq (Seed 4)** | **Boolq (Seed 5)** |
> |----------------|--------------------|--------------------|--------------------|--------------------|--------------------|
> | 2              | 82.46              | 82.47              | 82.45              | 82.46              | 82.45              |
> | 4              | 82.45              | 82.46              | 82.44              | 82.45              | 82.44              |
> | 8              | 82.44              | 82.45              | 82.43              | 82.44              | 82.43              |
> | 16             | 82.45              | 82.46              | 82.44              | 82.45              | 82.44              |
> | 32             | 82.40              | 82.41              | 82.39              | 82.40              | 82.39              |
>
> To summarize, IGU-LoRA does *not* assume identical gradients or scores for different samples. Each mini-batch contributes a distinct stochastic IG estimate, which is aggregated across batches and epochs. Theoretical results (Theorems 1 and 2) plus the above empirical evidence jointly show that the resulting importance scores are robust to reasonable variations in sample order and batch size.

---

> ### Author Response · Authors · 2025-11-28
> **Follow-up on Reviewer 4NAn's Feedback**
>
> Dear Reviewer 4NAn,
>
> I hope this message finds you well.
>
> I would like to sincerely thank you for taking the time to review our paper. Your valuable feedback is crucial in helping us improve the quality of our work.
>
> We have carefully **addressed all the concerns** you raised and made corresponding adjustments and **improvements in the revised version.** We hope that these changes adequately address your previous questions and contribute to further enhancing the paper's quality.
>
> In order to ensure that the paper can be timely revised within the ICLR timeline, we kindly request your prompt feedback. Your timely response will be vital for the next steps in refining the paper, and we look forward to your guidance to ensure the paper reaches its optimal quality in the final submission.
>
> Thank you once again for reviewing our work, and we eagerly await your timely reply.
>
> Best regards,
>
> Authors of  submission1904

---

### Official Review · Reviewer_bPkZ · 2025-10-31

**Soundness:** 2
**Presentation:** 2
**Contribution:** 3
**Rating:** 4
**Confidence:** 4

**Summary:**

This paper proposes IGU-LoRA, an adaptive rank allocation method for PEFT of LLMs. It integrates Integrated Gradients (IG) to compute within-layer parameter sensitivities and introduces an uncertainty-aware scoring mechanism based on exponential moving averages. The approach aims to overcome the instability and bias of gradient-based rank allocation methods such as AdaLoRA. Theoretical results include an error bound on the trapezoidal approximation of IG under a Hessian-Lipschitz condition. Empirically, IGU-LoRA shows consistent improvements over LoRA, AdaLoRA, and DoRA across GLUE, BoolQ, ARC, and GSM8K benchmarks.

**Strengths:**

1. The paper introduces a new importance-scoring mechanism that replaces instantaneous gradient magnitudes with parameter-space IG.
2. The experimental evaluation is comprehensive, covering multiple model scales (RoBERTa-large, Qwen-2.5-0.5B, Llama-2/3, DeepSeek) and diverse benchmark types.

**Weaknesses:**

1. In the example of Figure 2(b), any two parameter curves with the same integrated area would yield identical importance scores under Eq. (4). Consequently, the IG formulation cannot distinguish between parameters that are early-important and those that are late-important along the training.

2. Missing Recent Baselines. The paper omits several recent and closely related adaptive-rank methods, such as GoRA [1] (gradient-driven adaptive rank adjustment) and SalientLoRA [2] (saliency-based rank allocation).

[1] Gora: Gradient-driven adaptive low rank adaptation. NeurIPS 2025.

[2] Unveiling LoRA Intrinsic Ranks via Salience Analysis. NeurIPS 2024.

3. Some hyperparameters, such as $\beta_1$ and $\beta_2$ in Eq. (10), are not explicitly reported. Moreover, although Theorem 2 provides a stability guarantee under the condition $t \ge t_{\mathrm{burn}}$, the paper does not explain how this condition is verified or ensured in practice during training.

4. Eq. (4) assumes a zero baseline $\mathbf{W}^{(0)} = 0$, but the physical meaning of a zero-weight baseline is unclear for pre-trained models whose parameters are already non-zero. Should the pre-trained weights instead serve as the baseline? Furthermore, a key assumption of IG is path independence; yet, in a non-convex loss landscape, the straight-line path from 0 to $\mathbf{W}$ may not be meaningful or optimal. The paper does not analyze how different integration paths (e.g., stochastic or learned trajectories) might affect the resulting importance estimates.

5. Insufficient justification for division-based SNR vs. AdaLoRA's multiplication: The paper uses $\mathrm{SNR} = \frac{\bar{s}_e}{\bar{U}} \quad (\text{Eq. 11})$, interpreting uncertainty as "noise" to be penalized. However, AdaLoRA uses multiplication $s = \bar{I} \times \bar{U}$, interpreting uncertainty as "task diversity" to be rewarded. Table 4 shows only 0.32% improvement over multiplication (57.99→58.31), suggesting the choice may not matter much for these tasks.

**Questions:**

1. The reproduced results for some baseline methods differ substantially from those reported in the original papers, despite using the same model architectures and datasets. It would be helpful if the appendix could include detailed evaluation settings, such as whether multiple-choice questions are treated as open-ended generation or evaluated by choice-probability ranking. In addition, the statement “Each task is run with 5 different random seeds, and we report the median test performance.” is unconventional. Since five independent runs are already conducted, it would be more informative to report the mean ± standard deviation, which is the common practice for measuring performance stability and variance across seeds.

2. How sensitive is the final performance to the number of mini-batches M used for importance estimation in Algorithm 1? Has a sensitivity analysis been conducted to quantify how varying M (e.g., smaller or larger batch groups) affects both the stability of the importance scores and downstream accuracy?

3. The contribution of this paper is mainly a new importance-score method, and I hope it really is very important. Could the authors perform an ablation experiment to verify the interpretability of the proposed importance scores? Specifically, if a parameter (or singular direction) with a very high IG-based importance score is forcibly assigned a rank of zero (i.e., removed), does the model performance drop sharply? Such an experiment would provide more direct evidence that the IG-derived importance values truly capture critical structural contributions.

---

> ### Author Response · Authors · 2025-11-22
> **Response to Weaknesses 1, 2, and 3**
>
> **W1:**
>
> **Response:**
>
> Thank you for your valuable comments on the Integrated Gradients (IG) formulation and the example in Figure 2(b). We acknowledge that the current IG formulation assigns the same importance score to parameters with the same integrated area under the gradient curve, without considering the timing of their importance during training. The goal of IG is to capture the cumulative importance of parameters throughout training, focusing on their overall contribution, which is why we did not emphasize the early or later stages in the paper.
>
> In practice, during the early stages of training, the gradient is unstable, causing fluctuations in the importance scores. However, these scores stabilize as training progresses, and the final importance scores reflect this stability. Therefore, there is no significant impact from the early or later stages. We also provide experimental validation in **Appendix G** of the revised manuscript, which confirms that the mean and variance of the importance scores stabilize over time.
>
> **W2:**
>
> **Response:**
>
> Thank you for your valuable comment regarding the omission of several recent related adaptive-rank methods, such as GoRA [1] and SalientLoRA [2]. We greatly appreciate you pointing this out, as these methods are indeed closely related to our work.  In the revised manuscript, we now (i) add GoRA [1] to both the related work and experiments (Table 2), and (ii) cite and discuss SalientLoRA [2] in the related work section. Unfortunately, SalientLoRA does not provide an official implementation at the time of rebuttal, and we were unable to reproduce it reliably within the available time and resources. We therefore restrict ourselves to a qualitative comparison with SalientLoRA in the text, and leave a full empirical comparison to future work. The comparison results are shown in the table below. The results show that IGU-LoRA outperforms GoRA:
> | **Method**         | **# Params** | **BoolQ (acc)** | **ARC-e (acc)** | **ARC-c (acc)** | **GSM8K (acc)** | **AQuA (acc)** | **Avg.** |
> |--------------------|--------------|-----------------|-----------------|-----------------|-----------------|----------------|----------|
> | **Full FT**        | 494.0M       | 81.74         | **74.82**       | 54.98         | **34.64**       | *48.72*         | *58.98*  |
> | **Housbly-Adapter**| 9.0M         | 78.36           | 71.04           | 53.26           | 28.67           | 42.85           | 54.84    |
> | **LoRA**           | 8.8M         | 78.94           | 72.78           | 54.38           | 31.42           | 45.33           | 56.57    |
> | **AdaLoRA**        | 8.9M         | 80.32           | 73.90           | 54.23           | 33.27           | 46.58           | 57.67    |
> | **GoRA**           | 8.8M         | 79.24       | 71.20      | 51.91       | 32.07       | 45.81       | 56.04|
> | **IGU-LoRA**      | 8.8M         | **82.45**       | *74.62*         | **55.67**       | *34.16*         | **48.93**       | **59.17**|
>
> **W3:**
>
> **Response:**
>
> Thank you for your comment. We will respond as follows:
> 1. **The settings of some hyperparameters：**
> In IGU-LoRA, the specific settings of some hyperparameters ($M, N, \beta_1, \beta_2$) in the experiments are provided in the appendix, as shown in the table below:
> | Hyperparameter | $M$  | $N$  | $\beta_1$ | $\beta_2$ |
> |----------------|------|------|-----------|-----------|
> | Value          | 16   | 20   | 0.85      | 0.85      |
> 2. **Verify the condition $t \geq t_{\text{burn}}$:**
>
> Thank you for your feedback. In this paper, $t_{\mathrm{burn}}$ refers to the number of initial epochs that are discarded due to large fluctuations and instability in the training process. To confirm $t_{\mathrm{burn}}$, we observe the statistical properties of $y_t$ during training. Early in the training process, $y_t$ may be noisy and unstable. We select $t_{\mathrm{burn}}$ based on whether the **mean**, **variance**, and **autocorrelation** of $y_t$ stabilize after a few epochs, indicating that the model has "stabilized" and the burn-in period has ended.
>
> We set $t_{\mathrm{burn}}$ through experimental observation, and provide empirical support in **Appendix G**, where we plot the mean and variance of $y_t$ over different stages of training to validate this approach.

---

> ### Author Response · Authors · 2025-11-22
> **Response to Weaknesses 4 and 5**
>
> **W4:**
>
> **Response:**
>
> Thank you for your valuable feedback. In our implementation, IG is applied **only to the LoRA update** ($\Delta \mathbf{W} = \mathbf{AB}$), while the pretrained backbone ($\mathbf{W}\_0$) is kept frozen. Concretely, Eq. (4) is defined over the path ($\alpha \mapsto \alpha,\Delta \mathbf{W}$) with baseline ($\Delta \mathbf{W}^{(0)} = 0$), which corresponds to the state of “no adapter” (i.e., using ($\mathbf{W}\_0$) alone). Thus, the baseline is not a zero *network*, but a zero *update* relative to the pretrained model.  To express it more clearly and consistently, we have changed $\mathbf{W}^{(0)}$ to $\Delta \mathbf{W}^{(0)}$ and $\mathbf{W}$ to $\Delta \mathbf{W}$ in the revised manuscript.
>
> From this perspective, using ($\Delta \mathbf{W}^{(0)} = 0$) is analogous to the standard IG choice of a **"no-information"** reference, and allows us to quantify how much each entry of ($\Delta \mathbf{W}$) contributes when moving from “no adaptation” to the final adapted model.
>
> We agree that alternative baselines (e.g., different reference updates or paths that more closely follow the actual optimization trajectory) may lead to different attributions. A systematic study of baseline choices in the context of parameter-space IG is an interesting direction that we plan to explore in future work, and we will clarify this limitation in the revised manuscript.
>
>
> **W5:**
>
> **Response:**
>
> Thank you for your valuable feedback regarding the choice between the division-based Signal-to-Noise Ratio ($\mathrm{SNR}$) and the multiplication approach used in AdaLoRA. We greatly appreciate your observations, and we would like to clarify the reasons behind our choice of using a division-based $\mathrm{SNR}$ and the interpretation of uncertainty.
>
> 1. **Rationale Behind Division-based $\mathrm{SNR}$:**
>    - In our paper, we define the Signal-to-Noise Ratio ($\mathrm{SNR}$) in Eq. (11), where we interpret **uncertainty** as "noise" that we wish to **penalize** during the training process.
>    - The rationale for this choice is that we believe **high uncertainty** (represented by $U$) implies lower **reliability** of the model's parameters. Therefore, during training, we aim to **reduce the model's dependence** on uncertain parameters.
>    - **Penalty mechanism:** By dividing by uncertainty, we are effectively applying a penalty. This discourages the model from overly relying on uncertain parameters, as higher uncertainty generally leads to less reliable decision-making.
>    - **Theoretical foundation:** This approach is grounded in the intuition that higher certainty leads to more reliable decision-making, and we aim to reward the model for focusing on more confident parameters. This is consistent with statistical or signal processing principles, and we believe this is one of the key innovations of our research.
>
> 2. **Contrast with AdaLoRA's Multiplication Approach:**
>    - We have provided a comparative display of the uncertainty combinations between IGU-LoRA and AdaLoRA in the following table.
>
> | **Method**    | **Approach**         | **Interpretation of Uncertainty** | **Penalty/Reward**      | **Key Focus**               | **Interpretability**               |
> |---------------|----------------------|-----------------------------------|-------------------------|-----------------------------|-------------------------------------|
> | **IGU-LoRA**  | Division-based       | Uncertainty as "noise"            | Penalize uncertainty    | Reduces dependence on uncertain parameters | Can be interpreted as Signal-to-Noise Ratio (SNR) |
> | **AdaLoRA**   | Multiplication-based | Uncertainty as "task diversity"   | Reward task diversity   | Encourages diversity in handling tasks     | Limited interpretability           |

---

> ### Author Response · Authors · 2025-11-22
> **Supplement for Weakness 5**
>
> **W5:**
>
> Here is the new addition for Weakness 5, continuing from the previous reply, starting from point 3. We appreciate your understanding.
>
> **Response:**
>
> 3. **Key differences:**
>    - IGU-LoRA penalizes uncertainty, whereas AdaLoRA’s approach embraces uncertainty as a beneficial aspect of task diversity.
>    - Both approaches are valid but reflect different views on how uncertainty should be treated in model training. While our method rewards model confidence by penalizing uncertain parameters, AdaLoRA treats uncertainty as a source of strength, encouraging the model to diversify its learning across tasks.
>
> 4. **Performance comparison:**
>    - Regarding the experimental results in Table 4, we acknowledge that the improvement ($0.32\%$) between the division-based $\mathrm{SNR}$ and the multiplication-based $\mathrm{SNR}$ is relatively small. We believe this may indicate that, for the specific tasks evaluated, penalizing uncertainty (via division) has a relatively small impact on performance, while encouraging task diversity (via multiplication) contributes more significantly to performance improvement. This also suggests that the two methods may be complementary in certain cases, and other factors (such as task complexity, model architecture, or hyperparameter tuning) may have a greater impact on performance improvement.
>
> I hope these answers will help clarify your concerns.

---

> ### Author Response · Authors · 2025-11-22
> **Response to Questions 1 and 2**
>
> **Q1:**
>
> **Response:**
>
> Thank you for your valuable feedback regarding the discrepancies between our baseline method replication and the original paper's results, as well as the points you raised regarding the evaluation settings and performance reporting.
>
> 1. **Discrepancy in Replication Results:**
>    Thank you for your feedback. During the replication of the baseline methods, differences in parameter settings (such as random seeds, batch size, etc.) as well as variations in hardware and software configurations led to discrepancies with the results reported in the original paper. We fully agree with your suggestion, and in the revised manuscript, we will include more detailed information about the experimental setup in the appendix to enhance transparency.
>
> 2. **Reporting Test Performance (Median vs. Mean ± Standard Deviation):**
>    We understand your concerns, as "mean ± standard deviation" is a commonly used method for reporting performance. However, in our experiments, we noticed that when selecting certain random seeds, the model's performance could exhibit extremely high or low scores. These extreme values had a significant impact on the mean, causing distortion. In such cases, using the "mean ± standard deviation" approach may lead to bias. To better reflect the performance of most experimental results, we have chosen to use the median as the evaluation metric, as it is more robust to extreme values and better represents the model's overall performance. Similarly, some papers also report the median, such as:
>    [1] ALoRA: Allocating Low-Rank Adaptation for Fine-tuning Large Language Models
>    [2] EXLM: Rethinking the Impact of [MASK] Tokens in Masked Language Models
>    [3] XLNet: Generalized Autoregressive Pretraining for Language Understanding
>    [4] Funnel-Transformer: Filtering out Sequential Redundancy for Efficient Language Processing
>
> I hope these answers will help clarify your concerns.
>
> **Q2:**
>
> **Response:**
>
> Thank you for your insightful question regarding the sensitivity of the final performance to the number of mini-batches (denoted as $M$) used in Algorithm 1. In response to this question, we provide the following reply:
> 1. **Sensitivity Experiment:**
>
> We acknowledge that the choice of mini-batch number $M$ can impact the stability of importance scores and the performance of downstream tasks. In our experiments, we conducted sensitivity analysis not only for $M$ but also for other key hyperparameters, including the number of discrete points $N$ (Eq. (5) and (6)), and the smoothing coefficients $\beta_1$ and $\beta_2$ (Eqs. (9) and (10)). The values of $M$ and $N$ were chosen from the sets {$4, 8, 16, 20, 24$}, and $\beta_1$ and $\beta_2$ were selected from {$0.1, 0.3, 0.5, 0.8, 0.9$}. When studying the sensitivity of a specific parameter, the other hyperparameters were fixed at their default values. The experimental results are shown in the table below, using the BoolQ and GSM8K datasets to fine-tune the Qwen2.5-0.5B model.
> | $M$    | Boolq | GSM8K | $N$    | Boolq | GSM8K |
> |------|-------|-------|------|-------|-------|
> | 4    | 80.55 | 30.24 | 4    | 82.02 | 33.83 |
> | 8    | 81.34 | 32.29 | 8    | 82.05 | 33.88 |
> | 16   | 82.14 | 33.95 | 16   | 82.16 | 33.92 |
> | 20   | 82.16 | 33.74 | 20   | 82.14 | 33.95 |
> | 24   | 81.28 | 33.53 | 24   | 82.03 | 33.79 |
>
> | $\beta_1$    | Boolq | GSM8K | $\beta_2$    | Boolq | GSM8K |
> |------|-------|-------|------|-------|-------|
> | 0.1    | 81.93 | 32.14 | 0.1    | 82.05 | 32.14 |
> | 0.3    | 82.05 | 33.87 | 0.3    | 81.99 | 33.64 |
> | 0.5   | 81.88 | 33.78 | 0.5   | 82.22 | 32.78 |
> | 0.8   | 82.14 | 33.98 | 0.8   | 82.14 | 33.95 |
> | 0.9   | 82.05 | 33.93 | 0.9   | 82.09 | 33.87 |
> 2. **Result Analysis:**
>
> The results show that for both $M$ and $N$, as their values increase, performance slightly improves, but after reaching a certain threshold, it plateaus or even decreases. This indicates that the fine-tuning performance is robust to changes in $M$ and $N$, and excessively increasing these parameters does not lead to significant improvements, but instead may result in performance saturation or even degradation. Similarly, for $\beta_1$ and $\beta_2$, performance changes within a narrow range, showing robustness to these hyperparameters. This suggests that our method does not rely on careful hyperparameter tuning, and that the default configuration ($M=16, N=20, \beta_1=\beta_2=0.85$) is a reasonable choice across tasks.

---

> ### Author Response · Authors · 2025-11-22
> **Response to Question 3**
>
> **Q3:**
>
> **Response:**
>
> Thank you for your thoughtful comment and suggestion. To validate the effectiveness and interpretability of the proposed importance scoring method (based on Integrated Gradients, or IG), we conducted a series of ablation experiments focusing on high-impact parameters identified by the IG scores. Specifically, we performed the following steps:
> 1. **Rank Assignment:**
>
> We computed the parameter importance scores using IGU-LoRA, and assigned corresponding ranks to different parameter modules based on these scores. The size of the module's rank directly reflects its level of importance, with higher ranks indicating greater importance/impact and lower ranks indicating the opposite.
>
> 2. **Ablation Study:**
>
> We then selected some high-rank and low-rank modules for removal or set their ranks to zero (i.e., removing their influence on the model) and evaluated their performance. The experiments were conducted by fine-tuning Qwen2.5-0.5B on the Boolq and GSM8K datasets. In all these experiments, the model is first fully fine-tuned with IGU-LoRA. We then ablate selected modules **without further retraining**, and evaluate the trained model. Thus, the observed performance drops directly reflect the contribution of these modules to the final solution, which is precisely what our IG-based importance scores aim to capture.The experimental results are shown in the table below(**Appendix D**):
> | **#** | **Module Removed** | **Rank** | **Boolq** |
> |:-----:|:------------------:|:--------:|:--------:|
> | **1** | L3_K               | 10       | 81.15 (-1.30) |
> | **2** | L10_V              | 10       | 81.12 (-1.33) |
> | **3** | L3_K / L10_V       | 10 / 10  | 80.44 (-2.01) |
> | **4** | L1_K               | 5        | _82.40 (-0.05)_ |
> | **5** | L3_V               | 5        | 82.35 (-0.10) |
> | **6** | L1_K / L3_V        | 5 / 5    | 82.30 (-0.15) |
> | **7** | -                  | -        | 82.45 |
>
> ---
>
> | **#** | **Module Removed** | **Rank** | **GSM8K** |
> |:-----:|:------------------:|:--------:|:--------:|
> | **1** | L22_Q              | 12       | 32.35 (-1.80) |
> | **2** | L17_K              | 11       | 32.42 (-1.73) |
> | **3** | L22_Q / L17_K      | 12 / 11  | 31.15 (-3.00) |
> | **4** | L8_Q               | 6        | 34.05 (-0.11) |
> | **5** | L6_K               | 6        | 34.01 (-0.15) |
> | **6** | L8_Q / L6_K        | 6 / 6    | _33.84 (-0.32)_ |
> | **7** | -                  | -        | 34.16 |
>
> 3. **Performance Drop:**
>
> As shown in our experiments, removing modules with high IG values (ranked 10, 11, or 12) leads to a significant performance degradation on the evaluation tasks (Boolq and GSM8K). For example, removing high-ranked modules (such as the Q module L22_Q at layer 22) caused a performance drop of up to $1.80$ points on GSM8K, and removing the L10_V module resulted in a performance drop of up to $1.33$ points on Boolq. This significant performance drop provides strong evidence that the IG importance scores effectively capture the critical structural contributions of these parameters.
>
> In contrast, when we removed modules with low ranks (ranked 5 or 6), the performance drop was much smaller (e.g., only $0.05$ to $0.15$ points), indicating that the model is less reliant on these parameters.

---

> ### Author Response · Authors · 2025-11-28
> **Follow-up on Reviewer bPkZ's Feedback**
>
> Dear Reviewer bPkZ,
>
> I hope this message finds you well.
>
> I am the author of the paper titled "IGU-LoRA: Adaptive Rank Allocation via Integrated Gradients and Uncertainty-Aware Scoring," with the submission ID: 1904. First, I would like to sincerely thank you for taking the time to review our paper. Your valuable feedback is crucial in helping us improve the quality of our work.
>
> We have carefully **addressed all the concerns** you raised and made corresponding adjustments and **improvements in the revised version.** We hope that these changes adequately address your previous questions and contribute to further enhancing the paper's quality.
>
> In order to ensure that the paper can be timely revised within the ICLR timeline, we kindly request your prompt feedback. Your timely response will be vital for the next steps in refining the paper, and we look forward to your guidance to ensure the paper reaches its optimal quality in the final submission.
>
> Thank you once again for reviewing our work, and we eagerly await your timely reply.
>
> Best regards,
>
> Authors of  submission1904

---

> > ### Comment · Reviewer_bPkZ · 2025-11-28
> >
> > Thank you for the detailed and thoughtful rebuttal. Your clarifications have addressed the key concerns I raised in my initial review, particularly regarding the technical details and experimental validation. In light of these responses, I decide to updated my evaluation to reflect the improved clarity.

---

> ### Author Response · Authors · 2025-11-28
> **Response to Reviewer BPKJ**
>
> Dear Reviewer BPKJ,
>
> Thank you for your positive response! We sincerely appreciate your invaluable feedback about our work.
>
> We understand that the score cannot be changed currently, and we fully respect the reviewing policy. However, we would be grateful if you could kindly clarify your current score, as it would help AC make a more informed decision.
>
> Best regards,
>
> Authors of submission1904

---

### Meta-Review · Area_Chair_JzHC · 2026-01-06

**Summary:**

This paper proposes IGU-LoRA, an adaptive rank allocation method for LoRA that uses parameter-space Integrated Gradients combined with an uncertainty-aware scoring mechanism to improve stability and effectiveness over gradient-based adaptive methods such as AdaLoRA. Reviewers raised concerns about the interpretation of the IG baseline and path dependence, sensitivity to stochastic training factors such as batch size and data order, missing or insufficiently documented baselines and hyperparameters, and the lack of validation on very large models.

In the rebuttal, the authors clarified that Integrated Gradients is applied only to the LoRA update parameters (ΔW), with the zero baseline corresponding to the pretrained model without adapters rather than a zero-initialized network. This resolves the main conceptual confusion around the baseline choice. Path dependence is acknowledged as an inherent limitation of Integrated Gradients in non-convex settings, but the method relies on relative importance ranking rather than absolute attribution. Concerns about stochasticity were addressed through theoretical clarification and additional experiments showing stable rank allocation and downstream performance across different seeds, batch sizes, and shuffling orders. The authors also added missing baselines, hyperparameter documentation, sensitivity analyses, and interpretability ablations demonstrating that removing high-importance modules leads to substantial performance drops.

While limitations remain, most notably the absence of experiments on models larger than 30B parameters, the rebuttal satisfactorily addressed the core technical concerns. The remaining issues primarily affect scope and generality rather than correctness, supporting a borderline acceptance.

**Reviewer Concerns:**

**Concerns largely addressed in the rebuttal**

Reviewer **bPkZ**
(W1) Limitation of Integrated Gradients in distinguishing early versus late importance. The authors clarified that the goal is cumulative importance rather than temporal attribution and provided additional empirical evidence showing stability of importance scores over training.
(W2) Missing related adaptive-rank baselines such as GoRA and SalientLoRA. GoRA was added to experiments and SalientLoRA was discussed with justification for exclusion due to lack of a reproducible implementation.
(W3) Missing hyperparameter details and unclear verification of the burn-in condition. The authors added explicit hyperparameter settings, sensitivity analyses, and empirical validation of the burn-in criterion.
(W4) Interpretation of the zero baseline and path dependence in Integrated Gradients. The authors clarified that IG is applied to the LoRA update parameters ΔW with a zero-update baseline corresponding to the pretrained model without adapters, and explicitly acknowledged path dependence as a limitation.
(W5) Justification of division-based SNR versus AdaLoRA’s multiplication-based formulation. The authors provided conceptual clarification and additional discussion, and acknowledged that empirical differences are modest.

Reviewer **4NAn**
(W2) Concern about O(N) gradient overhead. The authors clarified that IGU-LoRA reuses standard training gradients via stochastic approximation and does not require additional backward passes, supported by efficiency experiments.
(W3) Sensitivity to batch size and data order. The authors provided theoretical justification and empirical results showing stable rank allocation and downstream performance across seeds, batch sizes, and shuffling orders.

Reviewer **8KC4**
Concerns about generalization and evaluation scope were partially addressed by adding MMLU results and multimodal experiments during rebuttal.

**Remaining limitations**

Reviewer **4NAn**, **8KC4**
(W1) Lack of direct experiments on very large models beyond 30B parameters. This limitation remains due to resource constraints and affects confidence in scalability, although the authors provided theoretical arguments and broader validation on models up to 8B.

Reviewer **bPkZ**
Residual concerns remain regarding the inherent path dependence of Integrated Gradients and the absence of a principled comparison across alternative integration paths. This is acknowledged by the authors as a limitation and does not affect the correctness of the proposed method but limits interpretability guarantees.

**Reviewer Scores:**

Reviewer bPkZ (Rating 4)
The rebuttal addressed the main technical questions (IG sampling sensitivity, batch size and shuffling order, and variance discussion), but this reviewer did not explicitly follow up. I would conservatively expect at most a small increase from 4 to 5.

Reviewer 8KC4 (Rating 6)
This reviewer explicitly stated they would raise their score after the rebuttal. Given the added MMLU and multimodal results plus fixes for clarity and reporting issues, I expect 6 to 7 or 8.

Reviewer FGvV (Rating 4)
No follow-up from the reviewer. While many questions were answered in the rebuttal, I assume the reviewer would keep the score at 4.

Reviewer oiw5 (Rating 8)
This reviewer already gave a strong accept and did not indicate outstanding blocking issues. I assume the reviewer would keep the score at 8.

Reviewer 4NAn (Rating 4)
No follow-up from the reviewer. The rebuttal provided detailed responses on overhead and robustness, but the lack of 30B+ experiments remains. I conservatively assume the reviewer would keep the score at 4.

Overall, based on explicit score updates and post-rebuttal comments, a reasonable post-discussion interpretation of the reviews is approximately 5, 8, 4, 8, and 4, corresponding to a borderline accept profile.

---

### Decision · Program_Chairs · 2026-01-26

Accept (Poster)